# The impact of low-mode symmetry on inertial fusion energy output in the burning plasma state

Indirect Drive Inertial Confinement Fusion Experiments on the National Ignition Facility (NIF) have achieved a burning plasma state with neutron yields exceeding 170 kJ, roughly 3 times the prior record and a necessary stage for igniting plasmas. The results are achieved despite multiple sources of degradations that lead to high variability in performance. Results shown here, for the first time, include an empirical correction factor for mode-2 asymmetry in the burning plasma regime in addition to previously determined corrections for radiative mix and mode-1. Analysis shows that including these three corrections alone accounts for the measured fusion performance variability in the two highest performing experimental campaigns on the NIF to within error. Here we quantify the performance sensitivity to mode-2 symmetry in the burning plasma regime and apply the results, in the form of an empirical correction to a 1D performance model. Furthermore, we find the sensitivity to mode-2 determined through a series of integrated 2D radiation hydrodynamic simulations to be consistent with the experimentally determined sensitivity only when including alpha-heating.

Harnessing and controlling fusion reactions for use as an energy source remains a major scientific objective with far-reaching potential. At the time of this writing, experiments at the National Ignition Facility (NIF) using the indirect-drive inertial confinement fusion (ICF) approach[1,2] have made significant progress in that pursuit[3–5]. These advances were directly enabled by using experimental observations to identify and mitigate degrading factors that reduce the rate of fusion self-heating and prevent ignition. Generalized ignition criteria often cast ignition in terms of an amplification of incident coupled energy, pressure, and yield[2,6,7]. The formalism developed in this work informs these criteria in terms of how compressional asymmetries and enhanced radiative loss impact the fusion yield amplification in a regime with significant fusion self-heating and directly relevant to ignition. These experiments initiate the Deuterium-Tritium fusion reaction $(D + T \rightarrow \alpha(3.5 MeV) + n(14 MeV))$ using the hot spot ignition concept. This concept relies on the conversion of kinetic energy from an imploding spherical shell, to form a hot dense core (or hot spot) of reacting DT at stagnation. In order to self-heat and burn the

surrounding DT fuel layer and achieve energy gain, $\alpha$-particle self-heating, $E_\alpha$, must exceed the compressional work, $E_{Pdv}$, done to create the hot spot[3]. A burning plasma is a critical stage required in all ignition experiments. The experiments described here are the first to reach this critical burning plasma state. In prior experimental work, necessary, but not sufficient conditions were determined and quantified[8–12]. In this work toward a burning plasma, we identify for the first time, the sufficient conditions for understanding the experimental fusion energy output. Notably, we build on the analytic model based on original work by Hurricane et al.[11] by including an empirical correction factor for mode-2 symmetry. This set of experiments is part of the same experimental campaign that has since achieved ignition[5] and gain[13,14].

The reacting hot spot, typically lasting from 80 to 150 ps, can be modeled based on energy sources and sinks. During this time, the energy in the hot core $E_{IE}$ can be described as $E_{IE} = E_\alpha + E_{Pdv} - E_{rad} - E_{cond}$[15]. Here $E_\alpha$, $E_{Pdv}$, $E_{rad}$, $E_{cond}$ are the energy deposited from alpha heating, energy from the compressional work,

✉ e-mail: ralph5@llnl.gov; ross36@llnl.gov; alex@pacificfusion.com

radiated energy from Bremsstrahlung, and energy conducted away from the hot spot through thermal transport respectively. A burning plasma state is reached when $E_a/E_{Pdv} > 1$[3]. To compare with experimental results, work by Hurricane et al.[11] provides a framework by solving a similar hot spot power balance expression for experimental conditions near the threshold of a burning plasma. This initial 1D framework has been augmented to correct for degraded fusion energy output resulting from mode-1 and radiative mix. The mode-1 correction accounts for the diversion of compressional energy $E_{Pdv}$ from mode-1 kinetic energy due to bulk hot spot motion[9]. The radiative mix correction quantifies the impact of increased Bremsstrahlung losses $E_{rad}$ from shell material entering the hot spot[8]. Including these terms is shown to improve agreement between model and experimental performance in smaller scale experiments dominated by radiative mix[8]. However, the same model over-predicts, by between 50% and 400%, the performance in 11 out of the 16 experiments in the campaigns included here that ultimately produced burning plasmas.

## Results

In this work, we include a correction factor for mode-2 asymmetry (also known as $P_2$) in the burning plasma regime. Comparing data with expectation, we find that this asymmetry alone has been sufficient to prevent the plasma from entering the burning state in the majority of experiments leading up to the burning plasma result[3]. With the inclusion of a correction factor for mode-2 using an empirical fit to data (and shown to be consistent in sensitivity to simulation results), the performance of all 16 experiments in this set can be explained within experimental error using the model. The high degree of correlation found by including these three corrections indicates that they are the dominant sources of shot-to-shot variation in performance (This data excludes experiments with very poor-quality DT ice layers and large surface particulates.). Furthermore, at these scales and with this design, sufficient alpha-heating is correlated with primary DT neutron yields exceeding $3 \times 10^{16}$, suggesting that asymmetry degraded performance to below-burning plasma levels in 44% of the experiments considered here[3]. Of the remaining experiments, 37% (6), succeeded in achieving a burning plasma. This would suggest that with optimized mode-2 symmetry, 81% of the implosions would have entered the burning plasma regime.

Eqn. (1) below is our model for expected fusion output and is based on the expression outlined by Hurricane et al.[11] for fusion yields in the approximate range of temperatures, 3 keV < $T_{ion}$ < 5 keV. Here we include correction factors for radiative mix[8] ($\eta_{mix}$), mode-1[9] ($\eta_{m1}$), and mode-2 ($\eta_{P_2}$):

$$Y_{mix,m1,P_2} \propto p_{abl}^{16/25} \frac{v_{imp}^{67/15}}{\alpha_{if}^{36/25}} S^{14/3} \eta_{mix} \eta_{m1} \eta_{P_2} \qquad (1)$$

The expected primary 14 MeV neutron yield (fusion energy output), $Y_{mix,m1,P_2}$ includes all three degradations according to the subscripts. The in-flight ablation pressure ($p_{abl}$) represents the drive pressure acting on the imploding shell ~500 ps before bang time (the time of peak gamma ray emission). The parameter, $v_{imp}$, is the implosion velocity. The in-flight 'adiabat', denoted by $\alpha_{if}$, is a measure of the entropy, or compressibility, of the DT fuel and is defined as the ratio of the fuel's pressure at peak velocity to its Fermi pressure. By design, the calculated adiabats, determined from radiation hydrodynamic simulations, for all these experiments were between 3 and 3.2[4]. Since there is, at present, no established method for inferring $\alpha_{if}$ directly from experimental observables alone, we assume it is constant for all subsequent analysis. The term $S$ is the scale factor of the implosion, taken to be the initial inner shell radius in micron $S \sim R_0$ scaled to 1000.

To compare to this model, we look at the experimental results of the two ICF experimental campaigns (I-Raum and HYBRID-E) that reached burning plasmas. See "Methods" section for full campaign details. These indirect drive ICF experimental campaigns use complex targets depicted in Fig. 1a, b. The targets consist of a thin spherical shell located in the center of a hohlraum (a high atomic number, Z, cavity), as shown in Fig. 1a which also shows the 5 μm diameter fill tube. To optimize efficient conversion from laser energy to x-ray energy, the hohlraums are made of gold-lined depleted uranium[16]. The HYBRID-E hohlraum is cylindrical with two 3.64 mm or 4 mm diameter laser entrance holes (LEHs) located on the top and bottom. The inner beams (shown in Fig. 1a in shades of green) deposit laser energy near the equator while the outer beams (shown in blue and pink) deposit energy near the LEHs. The I-Raum is similar, but has two regions along z with larger radii where the outer beams (shown in Fig. 1b) deposit their laser energy in the interior of the hohlraum. The four beams shown entering the hohlraums represent the approximate directions of the 96 beams entering through each (upper and lower) LEH.

The spherical high-density carbon (HDC) shell is filled with a 50/50 mixture of deuterium and tritium and cooled to 18.7 K to form a thin DT ice (fuel) layer as shown in Fig. 1b surrounding a core of DT gas. An initial shell inner radius of 1.0 mm is used for all I-Raum experiments, initial inner radii of 1.05 and 1.1 mm are used in HYBRID-E experiments. A comparison of the three rise laser pulses for the two campaigns are

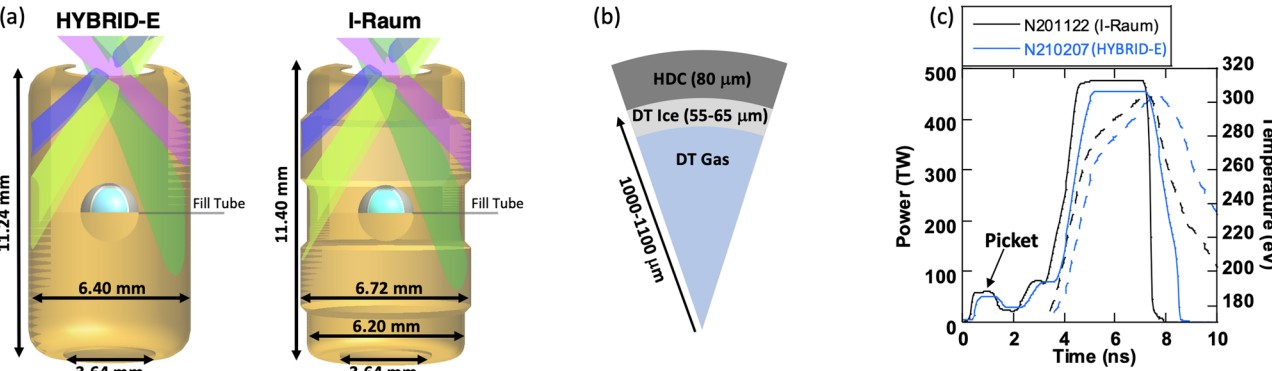

**Fig. 1 | Representations of that hohlraums, capsules and laser pulse shapes used in these experimental campaigns. a** The experimental setup is shown for the HYBRID-E (left) cylinder and I-Raum (right). The NIF lasers are grouped into four cones at angles relative to the axis, a single beam from each cone angle is shown for illustration. The inners are shown in shades of green 23° and 30° and the outer cones at 44° and 50° are shown in purple and blue, respectively. Half of the 192 beams enter the hohlraum through a laser entrance hole at the top of the target and half through the bottom. **b** Wedge schematic of the capsule, showing the features versus radius. **c** Two representative laser beam pulse shapes (solid lines) for the HYBRID-E and I-Raum experiments are shown and compared to the measured radiation temperatures (dashed lines).

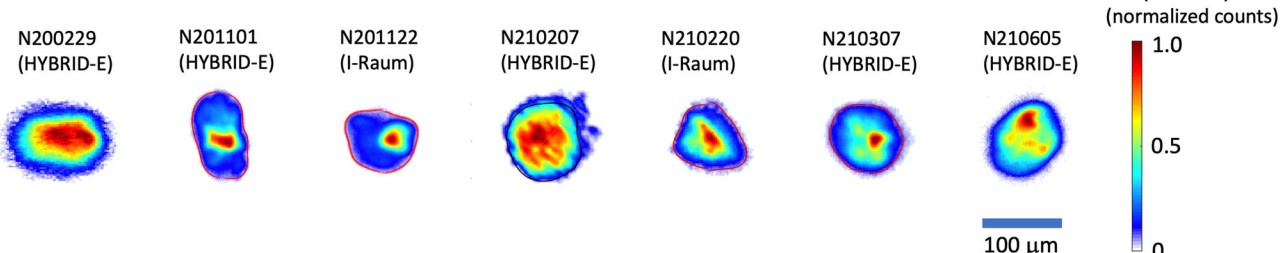

**Fig. 2 | Time integrated x-ray images of the "hot spot" where we infer $P_2$ symmetry and mix fraction.** N200229 (left) is an example of an oblate implosion with a P2/P0 (%) = −30.6 ± 2.5. The remaining experiments show the images of hot spot shapes from the experiments that achieved burning plasmas.

shown along with hohlraum radiation temperature time histories as measured by DANTE[17] in Fig. 1c. The shell is ablated by the hohlraum x-rays causing radial inward acceleration compressing the DT fuel and gas. During the implosion, preheat of the DT is reduced by using the three-rise pulse (shown in Fig. 1c) to create a series of three successive shocks that coalesce during the implosion near the DT ice/gas boundary. This allows for high compression and efficient transfer of energy to the hot core[4]. The shell sizes used in these experiments represent a 10 to 15% increase in scale, $S$, relative to prior campaigns[18,19], raising the energy absorbed in the shell by 30 to 75% in pursuit of a burning plasma. Because larger radii shells require additional thickness, to maintain the desired shock timing, in-flight ablation pressure, and compression, the duration of the laser pulse (x-ray pulse) is lengthened[10,18] by between 100 and 300 ps.

Extending the pulse duration and increasing the shell size creates challenges for maintaining implosion symmetry. This challenge stems from the reduced transmission of the inner beams to the wall near the equator of the shell (see Fig. 1) during the peak power portion of the laser pulse. This reduced transmission arises because the picket of the laser pulse (shown in Fig. 1), required to launch a first shock sufficient to melt the HDC, also produces an inwardly expanding plume (bubble) of gold plasma, originating at the location where the outer beams meet the interior of the hohlraum. This inwardly expanding bubble absorbs the inner beams[20,21] during the peak of the laser pulse, reducing transmission to the waist by between 40 and 75%. Without corrective measures, this inner absorption significantly reduces x-ray fluence on the capsule from the waist leading to an oblate implosion. In addition to reduced transmission of the inner beams, crossed-beam energy transfer (CBET) resulting from the crossing of the high-intensity beams in the plasma flowing out of the laser entrance holes principally transfers energy from the inner beams to the outer beams[22] and usually increases with pulse duration. The resulting inner beam attenuation from these processes leads to a mode-2 asymmetry along the axis of the hohlraum.

Both HYBRID-E and I-Raum use 0.3 mg/cm³ of helium in the hohlraum to reduce the wall bubble growth rate, while still staying below the threshold for laser-plasma-interactions[23]. Besides the helium fill, the two experimental campaigns use different approaches to compensate for the bubble absorption of the inner beams. HYBRID-E principally uses wavelength separation, ($\Delta\lambda = \lambda_{inner} - \lambda_{outer}$) to control the exchange of energy that occurs via CBET[22] between the outer and inner cones (see Fig. 1). Here $\lambda_{inner}$ and $\lambda_{outer}$ are the inner and outer wavelengths of the laser prior to frequency tripling. Changing $\Delta\lambda$ allows us to compensate for the inner beam absorption in the bubble and the flow-induced CBET by increasing the energy of the inner cone without reducing the total laser energy[4].

The I-Raum employs beneficial changes to the shape of the hohlraum to improve symmetry. This novel hohlraum geometry produces symmetric implosions with little or no use of CBET($\Delta\lambda$)[4,24,25] by reducing the inner beam absorption by the wall bubble. To do this, the I-Raum hohlraum uses "pockets" (regions of larger radius) where the

outer beam cones heat the wall. These 3.36 mm radius pockets, compared to a HYBRID-E cylindrical hohlraum with a constant 3.2 mm radius, allow more time (~150 ps) before the expanding high-Z bubbles intersect the inner beams[20]. Therefore, for a given pulse length and picket laser energy, this radial displacement increases x-ray drive at the waist during the peak leading to a more prolate implosion. By additionally reducing the radius near the equator to 3.1 mm (smaller than the HYBRID-E experiments described here), the I-Raum is able to maintain roughly the same wall area and hohlraum radiation temperature ($T_r$) as HYBRID-E. This is because $T_r \sim (E_{Laser}/A_{Hohl})^{0.32}$. Here $E_{Laser}$ is the total energy entering the hohlraum, and $A_{Hohl}$ is the hohlraum wall area. Note the I-Raum does benefit slightly from the smaller 1 mm inner radius HDC shell since $P_2 \sim \sqrt{R_{cap}}$ for a given hohlraum radius, where $R_{cap}$ is the outer radius of the HDC shell[26].

In either design, inadequate compensation for the diminished inner cone propagation results in oblate implosions $\frac{P_2}{P_0} < 0$, while overcompensation results in prolate implosions, $\frac{P_2}{P_0} > 0$. We quantify this asymmetry $\frac{P_2}{P_0}$ using the amplitude of the Legendre polynomial, $P_2$[26] of the 17% contour of the final implosion (hot spot) self-emission x-rays normalized to $P_0$. $P_0$ is the coefficient of the 0-order Legendre polynomial, the average radius of the hot-core self-emission along the imaging direction. In experiments, $1 \le \Delta\lambda \le 1.55$ Å was required to minimize $P_2$ in HYBRID-E designs and 0.5 Å $\Delta\lambda$ was required in I-Raum experiments mostly to negate the flow-induced CBET. Figure 2 shows x-ray images of the self-emission from the six experiments that achieved a burning plasma in addition to an example of an oblate implosion (N200229) that did not achieve a burning plasma. The experiment, N201101 is an example of a prolate implosion with a $P_2/P_0$ of 40.1 ± 4.8%, indicating overcompensation for the diminished inner cone propagation, using $\Delta\lambda = 1.75$ Å. The self-emission x-ray images of the remaining experiments show a variety of implosion shapes with small but measurable $P_2$ asymmetries.

The bright (dark red) regions seen in the images in Fig. 2 indicate regions where local "mix" is radiating and cooling the hot spot during stagnation. Higher atomic number carbon and tungsten doped carbon ablator material that is mixed into the hot spot is much more emissive than the DT plasma and enhances the radiative loss leading to significant changes in yield. While there is always some amount of the power radiated from Bremsstrahlung, the low-Z of the DT gas minimizes radiation losses. The total radiated power is given by, $P_{rad} = (P_{DT} + P_{mix}) \propto ne_{DT}(ni_{DT} + Z^2ni_{mix})$, where $Z$ is the atomic number of the hot material, $P_{DT}$ is the power radiated due to the hot DT, and $P_{mix}$ is the power radiated from mix[15]. Here $ne_{DT}$, $ni_{DT}$, and $ni_{mix}$ are the DT electron density, DT ion density, and mix ion density, respectively. Even for a perfectly clean DT hot spot, $P_{mix} = 0$, the radiated Bremsstrahlung power is higher than the power from fusion alpha heating for ion temperatures below ~ 4.3 keV. Our HDC shell uses tungsten (Z = 74) to reduce ablator decompression from preheating, however this material also radiates significantly when it enters the hot core through instability growth from individual seeds due to surface particles[27], shell

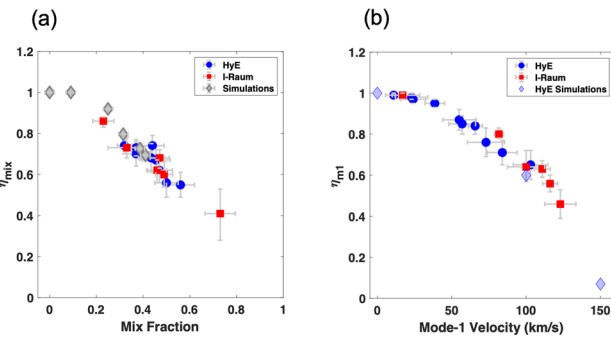
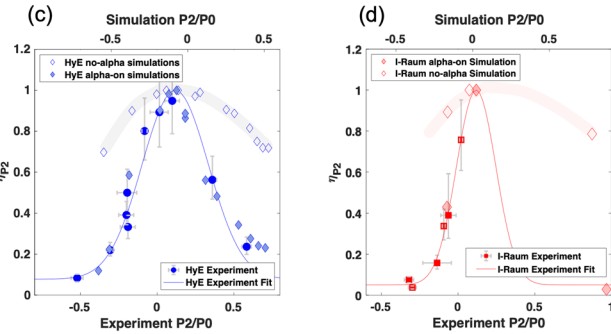

**Fig. 3 | Plots showing expected and inferred performance degradation. a** $\eta_{mix}$ the radiative mix degradation is plotted as a function of the measured fraction of x-ray emission from isolated mix. **b** $\eta_{m1}$, the mode-1 degradation is plotted as a function of the measured mode-1 velocity for the two campaigns. Simulations of the expected mode-1 degradation from simulations are shown for comparison. **c** $\eta_{P2}$ for the HYBRID-E design from experimental data using Eqn. 4 is plotted vs. $P_2/P_0$ as filled blue circles, and the experimental fit from Eqn. 5 is shown as the solid blue curve. Simulation results of $P_2$ only scans are plotted for comparison with experimental data. Simulation results including alpha heating effects are plotted as gray filled diamonds. Simulation scan results excluding alpha-heating effects are plotted as unfilled diamonds and the trend is highlighted with the thick light blue curve. To compare the sensitivity between the simulations and experiments, the $P_2/P_0$ axes for the simulations results (labeled above the plot) have required shifting relative to the experimental $P_2/P_0$ axes (shown below) because of a systematic

offset. (**d**)$\eta_{P2}$ for the I-Raum design from experimental data using Eqn. 4 (normalized so the maximum of the fit using Eqn. 5 is one) is plotted vs. $P_2/P_0$ as filled red squares, and the experimental fit from Eqn. 5 is shown as the solid red curve. Simulation results of $P_2$ only scans are plotted for comparison with experimental data. Simulation results including alpha heating effects are plotted as red filled diamonds. Simulation scan results excluding alpha-heating effects are plotted as red unfilled diamonds and the trend is highlighted with the thick light red curve. Again, to compare the sensitivity between the simulations and experiments, the $P_2/P_0$ axes for the simulations results (labeled above the plot) have required shifting relative to the experimental $P_2/P_0$ axes (shown below) because of a systematic offset. Error bars in the y-direction ($\eta$) represent the accumulated 1σ error from the calculations using analysis from multiple diagnostics. Error in the x-direction is error from analysis of a single diagnostic. Note: HyE indicates HYBRID-E in legends.

defects[28], or engineering features like the fill tube[8] or tent support[29,30]. As a result of these isolated sources, we can infer the fraction of x-ray radiation from mix by isolating the high spatial frequency bright features visible in the x-ray images as shown in Fig. 2. The very bright feature associated with mix due to instability seeded by the (5 μm) fill tube is clearly visible on the right-hand side of many of the images. Besides the fill tube, features are in-part due to small pits (typically smaller than 10 μm³) on the surface of the shell and voids within the shell (typically smaller than 40 μm³). Additionally, we can infer mix by correlating the total x-ray brightness with the neutron yield[31] (not shown). To determine the impact on performance, we follow work by Pak et al.[8],

$$\eta_{mix} = (1 - \beta \tau_{BW} T_{ion} F_{mix})^{0.93} \qquad (2)$$

The term, $\beta$ is a calibration constant established from historical data[8]. The gamma reaction history detector (GRH) measures gamma rays from the fusion reactions and produces a power time history. The full-width at half-maximum of this signal is the fusion burn width $\tau_{BW}$. $T_{ion}$ is the Brysk DT ion temperature determined from analysis of the Doppler broadened neutron time-of-flight spectra. The mix fraction $F_{mix}$ is the fraction of bright x-ray emission from the high spatial frequency "mix" features seen in hot core images[8,32] with respect to the total emission, as shown in Fig. 2 coming primarily from the fill tube. Measurements are sensitive to photon energies of >10 keV. We infer the fraction of the total radiative loss at photon energies >2 keV and find it is typically less than 10%.

The impact of radiative losses on yield $\eta_{mix}$ is plotted in Fig. 3a. Mixing occurs in both I-Raum and HYBRID-E data sets with the fill tube being a constant source for radiative mixing for both campaigns contributing between 10% up to 40% of the total x-ray emission (mix fraction). Here we compare the analysis of experimental data with analysis from simulated mix data[8]. The total degradation from radiative mix is found to be less than a 2× for the majority of the experiments. The plots in Fig. 3 show how mix, mode-1 and P₂ have affected the implosions. Note the error from each diagnostic and analysis method is allowed to propagate using the Monte Carlo error propagation method. The error bars presented everywhere in this work

represent 1σ (or 68% confidence). Data from the three HYBRID-E experiments with very poor-quality DT ice layers have been omitted from all analysis presented in this article; these are defined as layers with an effective groove RMS (K) exceeding 1 μm and with the largest groove area exceeding 300 μm². One additional implosion, found to be severely perturbed as a result of coalescing jets of mix correlated with particles found on the surface prior to the implosion is also omitted. All other DT implosion data including all I-Raum data is included.

In experiments, in addition to compressional mode-2 asymmetries that degrade performance and lead to spheroidal hot spot emission, there is also an inherent mode-1 compressional asymmetry. The magnitude and direction of the mode-1 asymmetry results from the summation of radiation asymmetries from the target window geometry and laser delivery together with the inherent asymmetry in shell thickness. This mode-1 can affect the shape of the hot spot, but mainly leads to a bulk motion[33] that can be accounted for by the amount of energy diverted from purely radial kinetic energy. This lost compressional energy substantially decreases the maximum achievable pressure and density[9]. Each shell is individually metrologized to ensure shell (mode-1) thickness uniformity meets our ignition specifications. In general, however, the inherent laser non-uniformity is not corrected. The exception in this data set was I-Raum experiment, N210220, with the goal to reduce mode-1 by tailoring of the 3D distribution of laser power to compensate for known mode-1s resulting from measured target and inherent laser nonuniformities. The mode-1 velocity ($v_{p1}$) is measured using time-of-flight detectors[34] which precisely measure the difference in arrival time between photons and the main neutron peak across five lines of sight to infer the bulk velocity of the neutron-emitting plasma. $v_{imp}$ is inferred from the bang time using a rocket model and relevant trajectory measurements on non-DT shots using the methodology described in refs. 35, 36. We account for mode-1 losses following the "piston" model[9].

$$\eta_{m1} = \left(1 - f^2\right)^{3.3} \frac{Y_{amp,3D}}{Y_{amp,1D}} \qquad (3)$$

Here the term, $f = v_{m1}/v_{imp}$ is the normalized mode-1 velocity as measured using the neutron time-of-flight (NTOF) spectrometers[37].

The term $\frac{Y_{amp,3D}}{Y_{amp,1D}}$, represents the reduction in yield amplification due to 3D imperfections. For this analysis, $\frac{Y_{amp,3D}}{Y_{amp,1D}} = e^{\left[-(2.3)\chi_\alpha^{1.2}f^2\right]}$, where $\chi_\alpha \sim P\tau_{BW}T_{ion}^{(1.3)}$. $P$ is the pressure[8,9].

In Fig. 3b, we plot $\eta_{m1}$ (Eqn. 3) along with the measured mode-1 velocities from the two campaigns. The experimental results are in agreement with hydrodynamic simulations (blue diamonds) of the HYBRID-E (1050 μm inner radius capsule) design. Overall, HYBRID-E experiments have mode-1velocities ranging from 11 to 103 μm/ns, indicating a range for $\eta_{m1}$ from 0.99 down to 0.65, respectively. The I-Raum range for mode-1 velocity is from 17 to 123 μm/ns, and the corresponding degradations $\eta_{m1}$ range is from 0.99 down to 0.46, respectively. The I-Raum, with an average $\eta_{m1}$ of 0.68 is losing more to mode-1 than Hybrid E with an average $\eta_{m1}$ of 0.86. This discrepancy has increased in the burning plasmas where the average $\eta_{m1}$ has dropped to 0.55 for the I-Raum, but remained roughly constant for Hybrid E.

To isolate the implosion performance sensitivity to $P_2$, we first estimate the performance from Eqn. (1) by including the results of the experimental analysis for both the 1D portion and the degradations from mix (Eqn. 2) and from mode-1 (Eqn. 3), leaving out $\eta_{P_2,j}$. We get two equations for $Y_{mix,m1,j}$, where the subscript $j$ indicates either the I-Raum or HYBRID-E data sets. Finally from Eqn. (1), setting $Y_{mix,m1,P_2} = Y_{DT}$, we solve for $\eta_{P_2 j}$ to get,

$$\eta_{P_2,j} = \frac{Y_{DT}}{Y_{mix,m1,j}} \quad (4)$$

In Fig. 3c, we show $\eta_{P_2}$ for HYBRID-E as a function of $P_2/P_0$. For these experiments, HYBRID-E prior to N210808, the degradation factor, $\eta_{P_2}$ is determined for each experiment from Eqn. (4). The lower x-axis is the hot spot x-ray $P_2/P_0$ for both the experimental data and for the empirical model which is fit to the experimental data in both Figs (c) and (d). We renormalize the HYBRID-E and I-Raum results so that empirical models described below and plotted in Fig. 3c, d, have a maximum of one. This provides useable degradation factors that can be compared with $\eta_{mix}$ and $\eta_{m1}$.

The HYBRID-E and I-Raum data, plotted in Fig. 3c, d, respectively, are fit to models of the sensitivity of neutron yield to $P_2/P_0$ using the following,

$$\eta_{P_2 j} \approx A_j e^{-\left[\frac{(P_2/P_0)-\delta}{W_j}\right]^2} + B_j \quad (5)$$

A Gaussian form, based on the shape of the sensitivity for $\eta_{P_2}$, although somewhat arbitrary, allows for a reasonable approximation with a limited number of terms for the present work. For HYBRID-E, Fig. 3c shows the fitted curves passing within the error bars of most of the experimental data. Here, a least squares method is used to find the terms $A_j$, $\delta$, $W_j$ and $B_j$ from the 10 Hybrid-E experiments. The subscript, $j$, is used to indicate individual constants for the I-Raum and HYBRID-E fits. $A_j$ is the amplitude of the gaussian portion. A minimum performance level, $B_j$, is allowed because we find measurable neutron yields in experiments, even with severe asymmetry. Because of this minimum, however, $W_j$ is not sufficient to discuss sensitivity to $P_2/P_0$. We use $\delta \sim 11\%$, the $P_2/P_0$ where the implosion is least degraded, to be based on the HYBRID-E data for both working models since we do not have prolate I-Raum implosion data, as shown in Fig. 3d. Without prolate I-Raum data, we also do not have a direct measure of the width for the case of the I-Raum. Alternatively we could have assumed $W_j$ to be the same for the 2 campaigns which would have shifted $\delta$ toward the prolate direction for the I-Raum case.

To compare the performance sensitivity to $P_2$ symmetry, we consider the $P_2/P_0$, where the yield is degraded by 50%. HYBRID-E analysis shows this at a $P_2/P_0$ of approximately ± 25% away from the

peak yield with $P_2/P_0$ of 11%. This is compared with ± 17% for the I-Raum implosions assuming the peak occurs in both cases at 11%. In addition, the trajectory of the fuel layer throughout the implosion is known to result in an assembled fuel layer at stagnation with an intrinsic low-mode thickness distribution. Minimizing asymmetry of the assembled fuel is an active area of research. For HYBRID-E, as the experimental shape diverges from $P_2/P_0 = 0.11$, the measured yield over the model decreases because the model over-predicts the yield without the inclusion of a $P_2$ degradation. This dependence is notable since it supports the cold fuel hypothesis that radiative mix, mode-1 and $P_2$ represent the primary sources degrading shot-to-shot performance.

The sensitivity of performance with respect to $P_2/P_0$ was further studied through a series of 2D radiation hydrodynamic simulations using HYDRA[38]. For HYBRID-E, these simulations were based on N210307[4] and were degraded with a fill tube mix in the hot spot. The simulations produced a series of $P_2$s by adjusting the laser cone fraction in fully integrated capsule/ hohlraum simulations. Each simulation was performed twice, once including the contribution from alpha heating and once without alpha heating. To map out the sensitivity with $P_2/P_0$, the simulated yields were normalized to the highest yield for the two cases (with and without alpha heating). The $P_2/P_0$ numbers from the simulations are produced from analysis of synthetic x-ray images similar to the x-ray image data analysis. For HYBRID-E, the results are plotted in filled diamonds with alpha heating on and in unfilled diamonds with alpha heating off in Fig. 3c (top x-axis) for comparison with data (blue circles). Three similar 2D simulations were conducted for the I-Raum (N210220) case and are shown in Fig. 3d. Note the I-Raum simulations do indicate that for very prolate implosions the performance is affected similar to HYBRID-E.

In both the HYBRID-E and the I-Raum cases, the simulation results (including the affect of alpha heating) shown in Fig. 3c, d are significantly more sensitive than the case without alpha heating. In HYBRID-E, for the case of no-alpha heating, increasing the $P_2/P_0$ by ± 50% results in only a correction factor of approximately 0.7. While the same ± 50% $P_2/P_0$ change results in a correction factor $\eta_{P_2}$ of between 0.1 and 0.2 for simulations with alpha heating. This is consistent with the 50% reduction in yield when increasing or decreasing $P_2/P_0$ by ± 25% from peak yield. Simulations of the I-Raum design show a similar difference between alpha-on and alpha-off simulations in Fig. 3d.

Both HYBRID-E and I-Raum simulations show an offset of the maximum $\eta_{P_2}$ with respect to our analysis of the experimental data. The HYBRID-E simulations show a maximum at −7.3% $P_2/P_0$ while the experiments show a maximum at 11%. I-Raum is also offset with our maximum simulation result occurring at 0.1% $P_2/P_0$. The exact reason for these offsets is not known, however, it may be caused by differences in the rate of change of $P_2$ ($dP_2/dt$), between the simulation and experiment. In addition, the absolute time history of the x-ray flux symmetry is not measured and differences may influence the resulting shape at stagnation. To compare sensitivity with $P_2/P_0$, the x-axis, "$P_2/P_0$ simulation", is shifted relative to "$P_2/P_0$ Experiment".

In Fig. 4a, a comparison of the model $Y_{mix,m1}$ ($\eta_{P_2} = 1$) to the experimental data, $Y_{DT}$, shows little correlation for most experiments. The gray line indicates one-to-one agreement between the model and the data. In Fig. 4b, $Y_{DT}$ is compared to the model $Y_{mix,m1,P_2}$. Here we have used the simple 1D model with all three correction factors, $\eta_{mix}$, $\eta_{m1}$, and $\eta_{P_2}$. The resulting plot with all data matching within error suggests that degradations from the three sources, mode-1, mode-2 and mix account for most of the shot-to-shot variability in measured performance. When comparing the plots, we can interpret the impact of $P_2$, noting that experimentally measured yields $\leq 2 \times 10^{16}$ do not enter the burning plasma regime with the threshold in these designs to be around $3.5 \times 10^{16}$, our lowest performing burning plasma. Of the 16 experiments shown in these plots, the 6 with the highest $Y_{DT}$ did achieve burning plasmas. Comparing the two plots, we find that of the remaining experiments, we would expect an optimized $P_2$ to have

improved the yield to $\geq 3.5 \times 10^{16}$ in 8 of the 10 remaining experiments, into the burning plasma regime.

## Discussion

In early 2021, the performance of DT layered implosions on the NIF was increased by a factor of $3\times$ in yield over previous results, into a new burning plasma regime[3] where alpha heating is now the dominant contributor to energy in the hot spot. Detailed analysis of experimental data accounting for mode-1 and radiative mix indicate that $P_2$ symmetry has played a major roll in degrading performance. Out of the 16 experiments considered in the study, 8 were degraded from $P_2$ by more than 50%. From this analysis, we find that these experiments would likely have entered the burning plasma regime, had the $P_2$ symmetry been optimized. This high sensitivity to $P_2$ is reinforced by simulation results which indicate that alpha heating, $E_\alpha$, greatly increases the sensitivity of $P_2$. Under burning plasma conditions alpha-heating heats the hot core more than compressional work ($E_\alpha > E_{PdV}$) leading to higher fusion energy output[3]. The data analysis and simulation results presented indicate that the rapid alpha heating process has been suppressed by degradation from asymmetric $P_2$. The rapid increase in performance that occurs as the implosion symmetry is improved with increased alpha heating significantly increases the performance sensitivity to $P_2$ in experiments otherwise meeting the threshold for burning plasma conditions. Prior to these experimental campaigns, our experience has been in the non-burning plasma regime where this level of sensitivity had not been observed[39]. Furthermore, while degradations were controlled to the level where they did not preclude burning plasmas, further reduction in enhanced radiation losses from high-$Z$ mix, and improvements in low-mode symmetry and higher modes of residual kinetic energy[40] are expected to additionally improve performance. Similarly, further increases in x-ray radiation coupling to the capsule, with a more efficient hohlraum or increased laser energy, are also projected to lead to continued performance improvements. As experiments improve, simulations do indicate that gains from $E_\alpha$ will increase faster than degradations and a larger fraction of the fuel will be burned up. As we make these improvements to the 1D parameters, the sensitivity of performance to $P_2/P_0$ and other degradations is expected to plateau as the fuel robustly burns[12].

## Methods

### Campaign summary

The HYBRID-E campaign originally began with larger radius (1100 µm) capsules, which achieved the previous record for implosion energetics and fusion yield (~56 kJ)[41]. The shots published in ref. 41 were limited to implosion velocities below ~370 km/s and had coast times ~1.3 ns. Longer coast times are deleterious because the imploding shell can begin to decompress in flight[10,42]. Initial attempts to increase the velocity of these implosions were limited by the quality of capsules fabricated to date at this scale[18] as well as symmetry. Relative to ref. 41, in this work we decreased the diameter of the laser entrance hole (LEH) to increase radiation drive on the capsule, and decreased the capsule radius by 50 µm. The first shot at these conditions, N201011, reproduced the velocity and coast time of N191110 from ref. 41 and had about half the fusion yield (~29 kJ), due to the smaller scale and from some mix from large capsule defects. The second experiment used an extended duration laser drive to simultaneously increase the implosion velocity to ~385 km/s and decrease the coast time to ~0.9 ns (increasing $p_{abl}$), this shot (N201101) produced a stagnation pressure roughly double that of its predecessor N201011 with a record yield, at the time, of ~100 kJ. With the extended pulse, the $P_2$ symmetry is expected to become more oblate, on N201101 this was compensated with $+0.5$ Å of $\Delta\lambda$ relative to N201011; this was an overcompensation due to model uncertainty resulting in a highly prolate shape on N201101. On the third experiment, N210207, this was reduced by 0.2 Å resulting in a $P_2$ shape much closer to round; a new capsule batch was

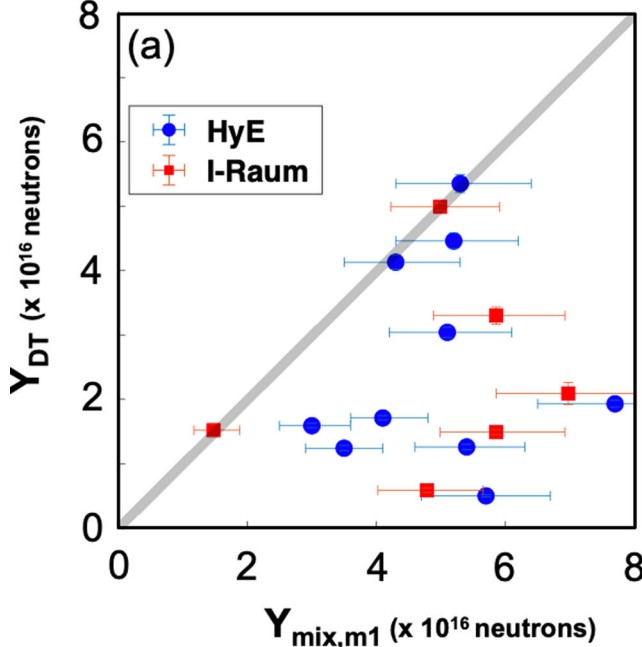

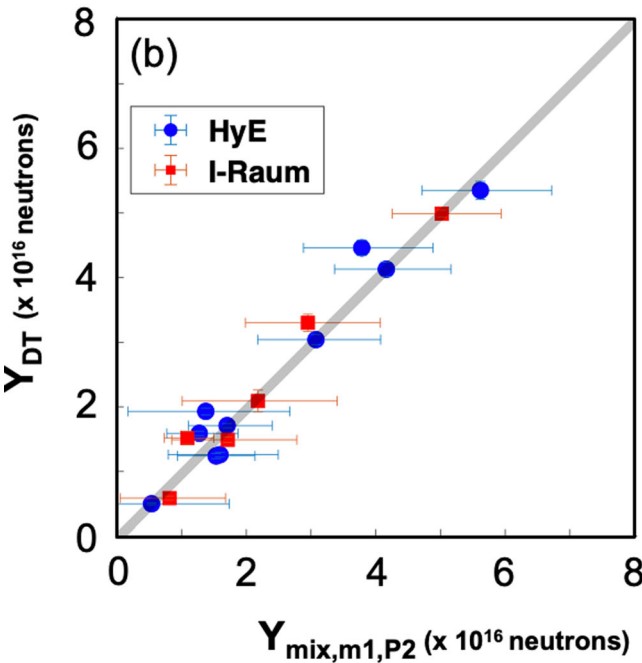

**Fig. 4 | Plots showing analyzed results.** In all plots the blue circles are from the Hybrid E campaign and red squares are from the I-Raum campaign. **a** The measured neutron yield ($y$-axis) is compared to the 1D model, Eqn. 1 degraded by radiative mix Eqn. 2 and mode-1 Eqn. 3 only. **b** The measured yield is shown compared to the 1D mode-l degraded by radiative mix, mode-1 and $P_2/P_0$. The gray curve shows the slope of 1. Error bars in the x-direction represent the accumulated $1\sigma$ error from using analysis from multiple diagnostics in determining results from Eqn. (1). Error bars in the y-direction represent uncertainty in our measurement of total DT yield.

also used on N210207, the velocity and coast time improved slightly due to laser delivery, and the combination of these changes resulted in a substantial increase in fusion yield to 170 kJ. The fourth experiment, N210307 is a repeat of experiment N201101 with a decrease in $\Delta\lambda$ from 1.75 Å to 1.55 Å resulting in a change in $P_2$ from $15.7 \pm 0.7$ µm (N201101) to $0.68 \pm 2.8$ µm (N210307). N210605 is a repeat of N210328 with a decrease in $\Delta\lambda$ from 1.4 to 1.0 Å. N210328 did not enter the burning

plasma regime, likely because it is the most prolate of any HYBRID-E experiment with $P_2$ of $23.7 \pm 1.2\mu$ (see Fig. 3c). Both N210328 and N210605 used a reduced ice thickness of 55 μm (compared with 65 μm) to increase the velocity relative to other recent burning plasma implosions.

The I-Raum campaign has used a single size of capsule, 1000 μm inner radius. The first experiment, N190217, used a capsule with very significant levels of defects (similar to the poor-quality capsules reported in ref. 28) resulting in high hot spot mix and low yield (~19 kJ), consistent with previous studies. The second DT experiment (N191105) used a better quality capsule and a higher initial shock pressure to increase the fuel adiabat and improve stability properties. N191105 observed a nearly tripled yield (50 kJ) but had a mild $P_2$ asymmetry and large mode-1 asymmetry, hypothesized to be from the loss of four inner beams. A repeated shot with full beam participation on N200816 reduced the mode-1 of N191105 but had a larger-than-expected negative $P_2$ asymmetry and resulted in a similar performance (50 kJ). The $P_2$ shape was tuned closer to round with the application of 0.5 Å of $\Delta\lambda$, leveraging the sensitivity curve demonstrated by HYBRID-E, on the fourth shot (N201122), with the yield more than doubling from its predecessor to ~106 kJ. Shot N201122 experienced a large mode-1 asymmetry, so on the next shot N210220 the incident laser power was adjusted to partially compensate for the known sources of mode-1, this shot produced a large increase in performance to ~160 kJ.

The absolute unscattered neutron yield is measured using neutron activation detectors (NAD)[43] and the magnetic recoil spectrometer (MRS)[44]; the final value is a weighted mean. The NAD measures neutron activation of three redundant zirconium samples to infer the yield based on the activity, known activation cross section, and sample mass and solid angle. The MRS measurement is independent and uses elastic scattering of deuterons in a foil close to the target, momentum analyzed using a dipole magnet; the systematic uncertainty is primarily the deuteron scattering cross section and an empirical calibration of the charged-particle collection efficiency. Each measurement also includes a statistical uncertainty, on these shots the two independent measurements agree with a typical difference of ~7%, consistent with the ~5% uncertainty on each measurement. $Y_{total}$ includes a correction for the fact that some fraction of the neutrons scatter in the dense shell[45], the ratio is $e^{4 \times DSR}$. This few percent uncertainty in yield propagates directly into all the burning-plasma criteria[3]. Key metrics and inferences from analysis of the data from the burning implosion experiments are included in Table 1.

## Gaussian Model for $\eta_{P_2}$
The parameter values used in the fit are given in Eqn. (5) and determined from nonlinear least square fit to the data for HYBRID-E are: 0.9225, 0.1180, 0.3058, 0.0775 for parameters, $A_j$, $\delta$, $W_j$, and $B_j$, respectively. The term $\delta$ is found from the HYBRID-E data and applied to both HYBRID-E and I-Raum. The remaining I-Raum parameters are: 0.9502, 0.1820, and 0.0498, corresponding to $A_j$, $W_j$, and $B_j$, respectively.

## Diagnostic methodology
$T_i$ is inferred from the 2nd moment of the neutron energy spectrum[46], measured using neutron time of flight (NTOF) detectors[37,47] which report an $T_i$ from both the DT and DD fusion reactions, the latter is less affected by residual kinetic energy (RKE)[48] in the burning fuel; the MRS also reports a DT $T_i$. Between 3–5 NTOFs at different lines of sight recorded good data for $T_{i,DD}$ and $T_{i,DT}$ on each shot and are self-consistent, the primary uncertainty results from the temporal instrument response function, which is measured using ultra-short impulse shots. Electron temperature measurements are not available on all shots but for N210207, the highest performer, a differentially-filtered measurement[32] gives $T_e = 5.19 \pm 0.14$ which is consistent with $T_{i,DD}$ as expected[49].

The mean hot-spot radius ($P_0$, $M_0$) and $P_2$ are measured with neutron[50] and x-ray[51] imaging; at the equator, these measure polar modes ($P_0$ and $P_2$) and from the pole, they measure the mean radius in the equatorial plane ($M_0$). In both cases, the measurement technique relies on either pinhole or penumbral imaging with the aperture placed close to the implosion. The x-ray detector is image plate with filtering that results in a measurement ~15 keV photon energy. The neutron imaging uses a redundant system of image plates and time-gated cameras to record images at different neutron energy ranges. Here we quote only $P_0$, $M_0$, and $P_2$, which are the dominant low-mode asymmetries measured in imaging diagnostics and the most important for understanding the shot performance and inferring a total volume. The measured x-ray volumes are systematically larger than neutron volumes, this offset is expected and depends on the photon energy predominantly measured in the imaging system. Since the hot-spot analysis methodologies[52] are validated using neutron-measured volumes these are used in the burning-plasma analysis in ref. 3.

Bang time (BT) is defined as the time of peak emission while burn width (BW) is defined as the full-width at half-max (FWHM) of the emission. Nuclear BT and BW are measured by the $\gamma$ reaction history instrument (GRH)[53], in which incident $\gamma$ rays Compton scatter electrons into a pressurized gas cell where they exceed the medium's speed of light, generate Cherenkov light, which is recorded by a photomultiplier tube. X-ray bang time and burn width are measured using SPIDER[54], which uses a set of differing attenuation filters with a slit and streak camera to record the x-ray emission history at varying photon energies. The temporal response of each instrument, and its absolute timing, is measured using laser-generated impulses and deconvolved from the measured signals to infer the BT and BW. These two measurements are very consistent with each other, especially in BT. The measured $BW_x$ may be systematically smaller due either to an instrumental effect or physics, as they measure differently-weighted burn histories and the x-ray history is known to be dependent on the photon energy; the largest discrepancy is the BW on N210207 in which the difference is a $1.2\sigma$ event with normally-distributed uncertainties, which is not unlikely (~11% likelihood).

DSR is calculated from the neutron spectra measured by NTOFs and MRS, and is defined as the ratio of the integrated neutron spectrum between 10–12 MeV and 13–15 MeV. The former represents neutrons which undergo a single elastic scattering in the compressed fuel, so the ratio is then related to the total areal density ($\rho R$) of the fuel[55]. Since each detector samples only a small fraction of the shell, and thus can be biased by 3-D asymmetries, a '$4\pi$' DSR, which is given in Table 1, is calculated by fitting a model to all of the available data (5 or 6 lines of sight depending on the specific experiment)[45].

The time-dependent $T_r$ is measured using the DANTE[17] instrument, which is an eighteen-channel x-ray spectrometer with a variety of calibrated transmission filters and x-ray mirrors coupled to x-ray diodes; $t_{coast}$ is taken as the difference in time between $T_r$ falling to 95% of its peak value and BT. Table 1 gives peak $T_r$ values and $T_r$ 500 ps before BT; DANTE did not acquire data for N210220. Lastly, $MF$ is a representation of the amount of x-ray emission due to high-$Z$ mix, or capsule material, that is injected into the hot spot during burn. High-$Z$ material enhances bremsstrahlung losses from the plasma, and $MF$ is extracted from the high-frequency components of the x-ray image data using the same prescription as in ref. 8. Note that this MF is for high-energy (>10 keV) radiation, an approximate relation to convert this into a bremsstrahlung power ($P_b$), where >2 keV photons are relevant, is $P_b \sim (1 + 0.4MF)P_{b,DT}$ (see Supplementary Material of ref. 8), where $P_{b,DT}$ would be the bremsstrahlung power for a pure DT plasma, so that a $MF \sim 0.4 - 0.5$ corresponds to a radiation loss enhancement of $0.16 - 0.20$. Previous work indicates that this mix arises from hydrodynamic growth of perturbations seeded by the capsule fill tube and capsule manufacturing defects[28].

**Table 1 | Key data and inferences for the six burning plasma experiments**

| | N201101 HYBRID-E | N201122 I-Raum | N210207 HYBRID-E | N210220 I-Raum | N210307 HYBRID-E | N210605 HYBRID-E |
|---|---|---|---|---|---|---|
| $Y_n$ (kJ) | 98.4 ± 2.7 | 106.1 ± 3.4 | 171.0 ± 4.8 | 160.6 ± 4.2 | 144.5 ± 3.9 | 131.6 ± 3.9 |
| $Y_{n,DT}$ (×10^{16}) | 3.49 ± 0.10 | 3.77 ± 0.12 | 6.07 ± 0.17 | 5.70 ± 0.15 | 5.13 ± 0.14 | 4.67 ± 0.14 |
| $T_{ion,DT}$ (keV) | 4.95 ± 0.12 | 5.17 ± 0.13 | 5.66 ± 0.13 | 5.54 ± 0.14 | 5.55 ± 0.11 | 5.89 ± 0.12 |
| $T_{ion,DD}$ (keV) | 4.62 ± 0.14 | 4.66 ± 0.14 | 5.23 ± 0.16 | 5.13 ± 0.24 | 4.92 ± 0.25 | 5.3 ± 0.21 |
| $4\pi$ DSR (%) | 3.44 ± 0.16 | 3.33 ± 0.14 | 3.16 ± 0.16 | 3.31 ± 0.14 | 3.48 ± 0.14 | 3.06 ± 0.14 |
| $\tau_{BW}$ (ps) | 141 ± 30 | 150 ± 20 | 137 ± 30 | 139 ± 20 | 138 ± 20 | 142 ± 20 |
| $t_{BT}$ (ns) | 9.37 ± 0.05 | 8.69 ± 0.05 | 9.13 ± 0.04 | 8.79 ± 0.04 | 9.33 ± 0.03 | 8.83 ± 0.03 |
| $t_{coast}$ (ns) | 1.22 ± 0.03 | 1.29 ± 0.05 | 0.99 ± 0.03 | 1.41 ± 0.04 | 1.16 ± 0.03 | 1.02 ± 0.03 |
| $T_{r,peak}$ (eV) | 301 ± 8 | 304 ± 8 | 307 ± 8 | NA | 306 ± 8 | 305 ± 8 |
| $T_{r,in-flight}$ (eV) | 270 ± 8 | 221 ± 9 | 283 ± 8 | NA | 276 ± 8 | 288 ± 8 |
| $v_{imp}$ (km/s) | 385 ± 12 | 380 ± 12 | 389 ± 12 | 374 ± 12 | 393 ± 12 | 394 ± 12 |
| $v_{m1}$ (km/s) | 55 ± 11 | 100 ± 12 | 73 ± 12 | 123 ± 10 | 57 ± 9 | 84 ± 10 |
| $P_0$ (µm) | 39.2 ± 2.6 | 39.0 ± 1.7 | 45.1 ± 2.3 | 41.9 ± 1.4 | 45.3 ± 3.7 | 46.3 ± 1.4 |
| $P_2$ (µm) | 15.7 ± 0.7 | 3.9 ± 1.2 | 4.5 ± 0.9 | −1.4 ± 0.8 | −3.7 ± 0.8 | 0.68 ± 2.8 |
| $F_{mix}$ | 0.44 ± 0.06 | 0.37 ± 0.04 | 0.47 ± 0.06 | 0.44 ± 0.05 | 0.47 ± 0.06 | 0.50 ± 0.05 |
| $\eta_{mix}$ | 0.68^{+0.07}/_{−0.07} | 0.62^{+0.06}/_{−0.05} | 0.7^{+0.06}/_{−0.06} | 0.73^{+0.05}/_{−0.05} | 0.62^{+0.06}/_{−0.06} | 0.56^{+0.06}/_{−0.07} |
| $\eta_{m1}$ | 0.87^{+0.05}/_{−0.05} | 0.64^{+0.08}/_{−0.07} | 0.76^{+0.07}/_{−0.07} | 0.46^{+0.07}/_{−0.07} | 0.85^{+0.05}/_{−0.04} | 0.71^{+0.06}/_{−0.06} |
| $\eta_{P_2}$ | 0.55 | 0.34 | 0.92 | 0.76 | 0.79 | 0.88 |

The implosion velocity is not directly measured but is scaled to experiments with direct measurements. We assume the velocity is proportional to the time of the rise-to-peak power to the time of peak x-ray emission, $t_{BT}$. We empirically assess $p_{abl}$ from DANTE measurements of the radiation temperature, with the ablation-pressure relationship[56] $p_{abl} \propto T_r^{3.5}$, which is related to coast time[10].

**Target quality**

Execution of a successful NIF experiment begins with fabrication[57] of the targets to be used, the design details are summarized in Fig. 1a and discussed in detail in ref. 4. The HYBRID-E target has a diameter of 6.40 mm and a length of 11.24 mm. The I-Raum target has a diameter of 6.72 mm at the outer beam location and a diameter of 6.20 mm at the inner beam location near the center of the target, with a total target length is 11.4 mm. The total hohlraum wall area is similar in both cases. The hohlraums are made of depleted uranium with a 0.7 µm gold overcoat, which are fabricated by deposition onto a mandrel, which can be diamond-turned to the precise shape for each hohlraum shape[58]. Each component is screened for coating defects that could lead to drive asymmetries or create particles that flake off of the wall onto the capsule. The laser beams enter the target through the laser entrance holes (LEHs), which are either 3.64 mm or 4 mm in diameter. The hohlraums include thin plastic windows covering the LEH to seal the interior volume and a gas fill line which is used to introduce the 0.3 mg/cc of helium gas.

The HDC capsule, located at the center of the hohlraum, includes a fractional atomic percentage tungsten-doped HDC layer that absorbs higher-than-thermal x-ray emission before it can "pre-heat" the fuel layer. The nominal total thickness (~80 µm, see Fig. 1b) and optical depth of the doped layer are specified by the capsule design to optimize the implosion adiabat and stability properties[4]. The capsules are fabricated using a chemical vapor deposition process[59]. After fabrication, the actual thickness (mass) of each layer is used to refine the pre-shot computational models[4]. The capsules are also all metrologized in depth to detect several kinds of defects that may be present, which allows a quality-control step to ensure the highest-quality capsules are used in the experiment, additional

quality control steps are completed after the completed target is assembled.

Several aspects of target quality can affect an experiment. For low-mode symmetry, the currently-dominant seed from fabrication is mode-1 asymmetry in the shell thickness[60], which can result from the capsule coating process. At present the thickness asymmetry is measured for each capsule before target builds by two x-ray transmission imaging[61] methodologies: contact radiography (CR), which measures the asymmetry from one view and is thus a random projection, and 'Xradia', which combines multiple views to reconstruct the full asymmetry magnitude. The uncertainty is similar in both cases and the two measurements are combined using a Bayesian model (BM) to create a more accurate estimation of the asymmetry amplitude. The orientation from these measurements to the final assembled target is unknown, so we additionally measure both the asymmetry's amplitude and its orientation when the target is installed in the cryogenic target positioner (Ctps) via the three orthogonal x-ray views[62]. A comparison of the of the measured mode-1 shell asymmetry amplitudes is shown in Table 2. The different imaging systems are in good agreement on the mode-1 amplitude. The fact that the shell used on N210220 had a substantial asymmetry, which was known before the experiment, motivated using a compensating laser asymmetry[63] to balance it.

The second category of target imperfections that are important for these experiments are defects or engineering features which can be a seed for instabilities that in turn injects ablator material into the hot spot. Engineering features include the fill tube[64] and capsule support tent[30], which are constant for all of these experiments. Randomly

**Table 2 | Measured capsule mode-1 asymmetry amplitudes via several complementary techniques**

| | N201101 | N201122 | N210207 | N210220 |
|---|---|---|---|---|
| CR (µm, ±0.2) | 0.27 | 0.44 | 0.10 | 0.55 |
| Xradia (µm, ±0.2) | 0.12 | 0.34 | 0.16 | 0.47 |
| BM (µm) | 0.22^{+0.14}_{−0.13} | 0.41^{+0.15}_{−0.15} | 0.18^{+0.14}_{−0.11} | 0.55^{+0.15}_{−0.15} |
| Ctps (µm, ±0.21) | 0.25 | 0.34 | 0.22 | 0.65 |

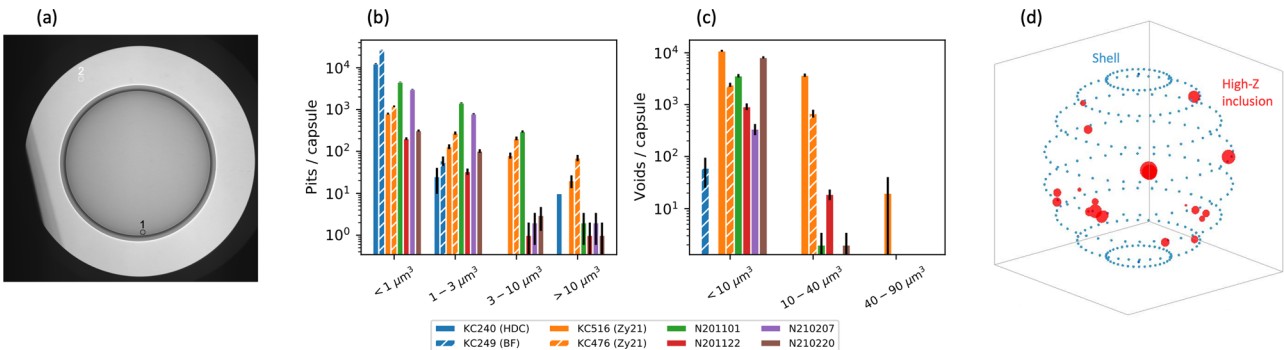

**Fig. 5 | Capsule metrology used to assess capsule quality with respect to seeds for instability that may lead to mixed mass.** Fig. **a** Pre-shot x-ray radiograph of the capsule used on N210207 through the LEH, with the capsule at center. Two particles were detected via a parallax analysis. Figs. (**b**) and (**c**) represent pits and voids per capsule, respectively, with several bins by volume for capsule batches used in previous work with good quality (KC240, KC249; blue) and marginal quality (KC516, KC476; orange). Fig (**d**) shows tomography-detected high-$Z$ inclusions on the shell used for N210207.

varying defects include 'particles', 'pits', 'voids', and 'high-$Z$ inclusions', which are discussed here.

'Particles' are usually flakes of high-$Z$ material, predominantly from the hohlraum wall, which delaminate and fall onto the outer surface of the capsule during fabrication[8]. The particle is then a seed for ablation-front instability growth, which can then inject high-$Z$ material into the fuel, which causes additional radiation losses. Particles are measured after the target is completed with x-ray imaging through the LEH with target displacements to identify, via parallax, whether a particular particle is on the capsule surface. The number and area of particles are thus characterized. An example image for the target used on N210207 is shown in Fig. 5a. In this case, there are two particles detected which are circled and labeled. In red, particle #1 is on the capsule and has an area of 79 μm². The second particle, #2, is on the lower LEH. Shots N201101 and N210220 had no particles while N210207 had the one mentioned earlier and N201122 had three (one with area 85 μm² and two with areas of 55 μm²). In general particles with areas < 100 μm² are considered relatively small, but their impact on radiative loss is not entirely understood and may be contributing to the inferred MF (e.g. in Table 1).

There are several categories of manufacturing defects in the capsules themselves. First, 'pits' are defects of missing material on the outer surface of the shell, equivalent to a divot. As soon as the capsule is driven, these are unstable to the ablative Rayleigh-Taylor instability. 'Voids' are regions of missing material within the bulk of the shell. A void can cause a seed for instability growth when the ablation front reaches it. Previous data have demonstrated that both are deleterious[28,41] for implosion performance. In Fig. 5b, c, we compare the pits and voids on the actual capsules used on these experiments to batches typical of previous experiments, such as the HDC[19] and BigFoot[39,65] (BF), which had good-quality capsules, as well as the shots reported in ref. 41 (Zy21), which were marginal quality. In previous work we had to rely on batch-averaged quantities due to average metrology methodology, here we have full-sphere measurements of the pit quantities and for the larger voids. Pits are now measured with a LynceeTec digital microscopy system, voids are measured using a x-ray computed tomography system. Larger voids are measured across the full sphere with a 4 × magnification tomography while the smaller voids (<10 μm³) are taken from a 20 × magnification tomography of ~5% of the shell.

We see that the smaller pits are reasonably comparable to capsules used in several previous works. Larger pits are not as good as the good-quality batches but are significantly improved from the marginal capsules used in ref. 41. Similar for voids, the smallest voids are comparable to previous batches while the larger voids are nearly

non-existent on these shells, like previous good-quality shells. Our general understanding is that it is the larger defects which inject the most mix material and the smaller defects, while numerous, are typically below a volumetric threshold at which they could inject material.

During this experimental series, an unusual defect was detected for the capsule batch used on N210207, which included high-$Z$ 'inclusions' deep in the shell, predominantly within the W-doped layer of the shell. These are likely Mo inclusions from components of the coating system that were improperly cleaned during that particular coating run. The impact of these defects on the shot is not understood at present; for these experiments, we believe that only N210207 was potentially affected. The defects are visualized using our tomography data in Fig. 5d.

## Cryogenic fuel layering

Several days before the shot itself, the target is installed onto a cryogenic target position (CryoTarpos) for the fueling and layering[62] process. Surrounding the hohlraum is a thermo-mechanical package connected to a cryostat, which enables exquisite control over the target temperature. Nominally equimolar DT fuel, with the actual composition measured beforehand via mass spectroscopy, flows into the capsule through a fill tube, where it is condensed into a liquid. Three orthogonal x-ray imagers measure the fuel inventory, and record the growth of a concentric spherical shell of DT ice on the inside of the ablator as the temperature is reduced below the triple point, to -18.6 K, and the fuel layer is formed via the 'beta layering' technique[66]. The quality of the ice layer is measured and compared to specifications. The HYBRID-E (I-Raum) uses 65 (55) μm thick ice layers. Once the ice layer is complete the target is inserted into the target chamber and aligned relative to the laser beams[67] to a tolerance in horizontal (vertical) position of ±11 (±15) μm and orientation angles, relative to the positioner, of ± 0.067°.

During the cryogenic layering the important quantities are first, to get the right inventory of fuel loaded, and second, to grow a high-quality concentric ice layer. Scalar quantities for the as-shot fuel composition and ice thickness are given later, in Table 3. As the layer is grown, three orthogonal x-ray imaging systems record data on the low-mode and high-mode shape of the ice. Example images from just before shot N210207 are shown in Fig. 6a. The images are each labeled at the top by a $\theta$ or $\phi$ value corresponding to the chamber coordinate system orientation, and are unwrapped to then be displayed as a function of the other angular coordinate. At $\theta = 90°$ the imaging system has a clear view of the layer through the LEH. At the equatorial plane patterns, called a 'starburst', are cut out of the hohlraum wall to enable imaging along orthogonal axes. The

location of the DT ice layer is marked. Feedback systems with the cryostat and localized heaters adjust temperature gradients within the target to produce a low-mode shape within spec, ±0.5 μm for modes 1–4.

The high-mode uniformity of the layer is also important, as these features can provide seeds for hydrodynamic instabilities. In particular, the layer is prone to forming grooves. A single scalar parameter $K$, with units of length, quantifying the RMS amplitude of these grooves was developed and published as Eq. 26 of Haan et al.[68] and is used, in combination with the area of the largest single defect, to characterize the high-mode layer quality. Specifications were developed for 'ignition' and 'tuning' specifications[68]; in terms of K these are 0.7 and 1.5 μm respectively and are 200 and 500 μm² for the largest defect. Figure 6b shows the layer quality metrics relative to these specifications for these shots, as well as all prior layered experiments conducted on NIF. The final as-shot layer quality is plotted, in these experiments, several attempts are typically required to obtain an adequate-quality ice layer. Only one burning plasma experiment, N210307 is clearly within the ignition specification.

### Table 3 | Summary of experimental configuration parameters

|         |                | N201101 | N201122 | N210207 | N210220 |
|---------|----------------|---------|---------|---------|---------|
| *Laser* | Energy (MJ)    | 1.89    | 1.82    | 1.93    | 1.78    |
|         | Power (TW)     | 490     | 485     | 470     | 480     |
|         | Cone Fraction  | 32.6    | 32.9    | 32.4    | 32.8    |
|         | Picket (kJ)    | 40.0    | 48.7    | 41.3    | 49.8    |
|         | Duration (ns)  | 8.15    | 7.4     | 8.05    | 7.4     |
|         | $\Delta\lambda$ (Å) | 1.75 | 0.5   | 1.55    | 0.5     |
| *Capsule* | IR (μm)      | 1049.2  | 999.9   | 1048.8  | 1000.0  |
|         | Mass (μg)      | 3905.7  | 3738.6  | 3881.5  | 3739.9  |
|         | W (%)          | 0.44    | 0.42    | 0.28    | 0.42    |
| *Fuel*  | DT Layer (μm)  | 64.7    | 55.6    | 64.2    | 55.3    |
|         | $f_D$ (%)      | 50.38   | 50.60   | 50.83   | 50.91   |
|         | $f_T$ (%)      | 49.41   | 49.19   | 48.96   | 48.88   |
|         | Age (hr)       | 128.9   | 175.8   | 163.5   | 126.1   |

## Laser delivery

The NIF laser system[69] generates the specified pulse shape, or power as a function of time, on each beam during the experiment. The pulse shape is designed to result in the desired radiation temperature history[4,70]. Wavelength settings, which then provide our CBET control, are controlled by the master oscillator[71]. The actual laser delivery is closely monitored to ensure it is sufficiently accurate compared to the request[72]. As the NIF optics are damaged on high-power experiments[73,74], the lifecycle is managed to ensure the best possible performance on these high-priority experiments. The laser settings are adjusted between shots to adjust the implosion physics or compensate for changes in the target configuration using pre-shot simulations[4]. The laser pulse, when incident on the target, produces a peak radiation temperature of just over 300 eV.

Example laser pulses were given, in total, in Fig. 1 and are also discussed in ref. 4. Accurate delivery of the requested laser pulse throughout its duration is key for performing high-quality experiments, as mis-delivery can affect the symmetry of the implosion - e.g. if the delivered picket energy, which launches the bubble, varies, or if the inner or outer cones deliver varying levels of energy, the $P_2$ symmetry can deviate from expectations. Inaccurate delivery can also contribute to the mode-1 asymmetry of the implosion[75]. The delivery accuracy can affects the implosion parameters, especially if overall energy or power is over- or under-delivered, which affects $v_{imp}$ and $t_{coast}$. In Fig. 7 we plot the actual as-delivered laser pulses for the cone averages compared to the request, plotted as actual power on the main axes and as a ratio of delivered/requested on the inset axes. In general we note that the delivery is quite good, although these experiments mostly have higher than requested outer-cone power during the peak, and N210207 experienced a noticeable picket over-delivery.

## Shot details

Table 3 gives several detailed as-shot parameters for the four experiments, grouped into categories of laser, capsule, and fuel parameters. The laser parameters include the as-delivered energy and power, cone fraction (defined as the inner power during the peak divided by the total power), picket energy, pulse duration, and Δλ. The capsule parameters are the key dimensions of inner radius, total mass, and dopant fraction. The fuel parameters include the final layer thickness (which, with the DT density ~0.25 g/cc, determines the total fuel mass),

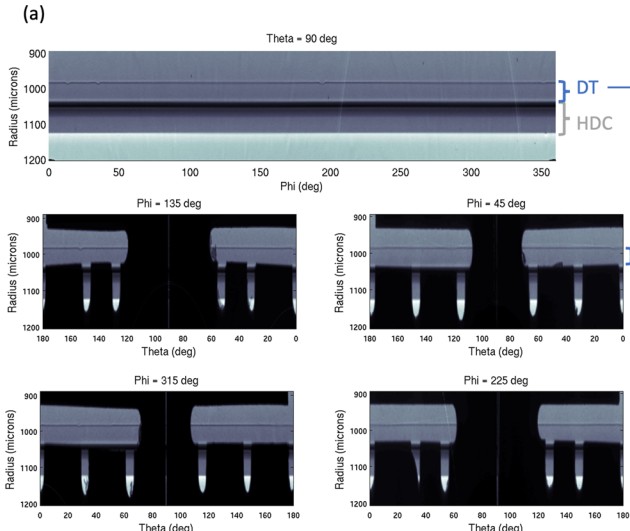

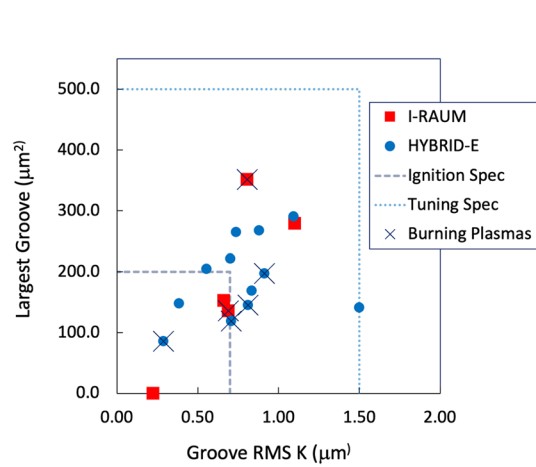

**Fig. 6 | Images used to analyze the DT layer and analysis of the DT layers from this data set.** Fig. (**a**) shows the unwrapped images from the cryogenic layering process for N210207 as an example of the images used to determine the layer quality. The plot in Fig. (**b**) shows the high-mode layer quality measurements for these NIF these specific experiments compared to the ignition and tuning quality specifications.

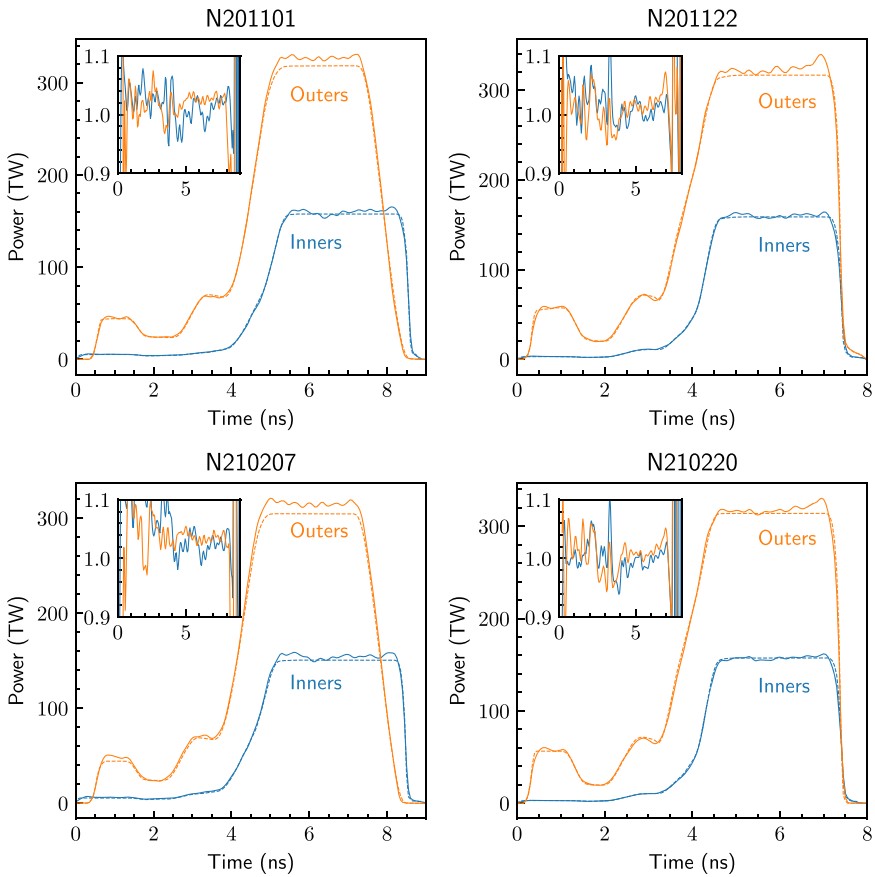

**Fig. 7 | Delivered laser pulses (solid lines) compared to the request (dashed lines) for inner (blue) and outer (orange) cones for all four shots.** The inset axes show the ratio of the delivered to requested power as a function of time.

the atomic fractions of deuterium and tritium ($f_D$ and $f_T$), and the age of the fuel. The latter is the duration since the last purge of He, which is key to avoid high levels of $^3$He accumulating in the capsule due to $\beta$ decay of tritium.

## Data availability
Raw data were generated at the National Ignition Facility and are not available to the general public. Derived data supporting the findings of this study are available from the corresponding authors upon request.

## Code availability
The simulation codes used in this manuscript are not available to the general public.

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

## Acknowledgements

We thank John Kline (LANL) and Mike Farrell (GA) for thoughtful discussions. The contributions of NIF operations and target fabrication teams to the success of these experiments are gratefully acknowledged. This work was performed under the auspices of the U.S. Department of Energy under Contract No. DE-AC52-07NA27344 and Contract no. 89233218CNA000001. This document was prepared as an account of work sponsored by an agency of the United States government. Neither the United States government nor Lawrence Livermore National Security, LLC, nor any of their employees makes any warranty, expressed or implied, or assumes any legal liability or responsibility for the accuracy, completeness, or usefulness of any information, apparatus, product, or process disclosed, or represents that its use would not infringe privately owned rights. Reference herein to any specific commercial product, process, or service by trade name, trademark, manufacturer, or otherwise does not necessarily constitute or imply its endorsement, recommendation, or favoring by the United States government or Lawrence Livermore National Security, LLC. The views and opinions of authors expressed herein do not necessarily state or reflect those of the United States government or Lawrence Livermore National Security, LLC, and shall not be used for advertising or product endorsement purposes. LLNL-JRNL-821520-DRAFT.

## Author contributions

J.E.R. contributing experimental physicist for both HYBRID-E and I-Raum campaigns, I-Raum experimental lead (following J.S. Ross), performed a detailed re-analysis of data for both campaigns, conducted a major revision of the manuscript; J.S.R. I-Raum experimental lead, and wrote sections of the paper; A.B.Z. hot-spot analysis lead, Hybrid-E experimental lead, wrote sections of the paper; A.L.K. Hybrid-E design lead, integrated hohlraum group lead; H.F.R. original I-Raum design lead; C.V.Y. present I-Raum design lead; O.A.H. capsule scale/burning plasma strategy, theory, 0D hot-spot models; D.A.C. empirical hohlraum $P_2$ model and hohlraum strategy; K.L.B. Hybrid Shot RI; D.T.C. Hybrid Shot RI; T.D. Hybrid Shot RI; L.D. 3D hot-spot analysis; M.H. Hybrid Shot RI; S.L.P. Hybrid Shot RI; A.P. Hybrid & I-Raum Shot RI, physics of capsule engineering defects; P.K.P. 1D hot-spot analysis, $Y_{amp}$ and GLC inference; R.T. Hybrid Shot RI; S.J.A. capsule microstructure physics; P.A.A. hohlraum physics; L.J.A. engineering and targets; B.B. penumbral x-ray diagnostic; D.B. computational physics; L.R.B. x-ray framing camera; L.B.H. HDC design and campaign lead; R.B. ICF physics/ignition theory; S.D.B. cryo layering; J.B. capsule fabrication; R.M.B. RTNAD nuclear diagnostic; N.W.B. neutron diagnostics; E.J.B. project engineering; D.K.B. diagnostics; T.B. capsule fab & metrology; T.M.B. cryo layering; M.W.B. project engineering; P.M.C. DT EOS measurements; B.C. HYDRA code development; T.C. LPI physics; H.C. GLEH x-ray diagnostic; C.C. target fab planning; A.R.C. ignition theory; D.S.C. capsule/instability physics; J.W.C. capsule fabrication; E.L.D. experiments; T.R.D. capsule physics; M.J.E. program management; W.A.F. hohlraum physics; J.E.F. 2DConA image analysis; D.F. nuclear diagnostics; J.F. magnetic recoil spectrometer nuclear diagnostic; J.G. ensemble simulations; M.G.J. magnetic recoil spectrometer diagnostic; S.H.G. ICF physics; G.P.G. nuclear diagnostics; S.H. capsule physics, iPOM analysis; K.D.H. neutron diagnostics; G.N.H. experiments; B.A.H. capsule physics; J.H. computational physics; E.H. nuclear time-of-flight diagnostics; J.E.H. MOR and PAM stability, SSD improvements, and FC control; V.J.H. MOR and PAM stability, SSD improvements, and FC control; H.W.H. gamma diagnostics; M.C.H. program management; D.E.H. hohlraum physics, CBET studies in Hybrid-C; D.D.H. capsule physics; J.P.H. x-ray diagnostics; W.W.H. management; H.H. capsule fabrication; K.D.H. ensemble simulations; N.I. x-ray diagnostics; L.J. x-ray diagnostics; J.J. neutron diagnostics; O.J. hohlraum physics; G.D.K. HYDRA code development; S.M.K. neutron diagnostics; S.F.K. x-ray diagnostics and analysis; J.K. diagnostic management; Y.K. gamma diagnostics; H.G.K. gamma diagnostics; V.G.K. neutron diagnostics; C.K. capsules; J.M.K. HYDRA code development; J.J.K. targets; M.K.G.K. ICF physics; B.K. ensemble simulations; O.L.L. velocity analysis; S.L. laser plasma instability (PF3D) code development; D.L. NIF facility management; N.C.L. optical diagnostics; J.D.L. ICF physics; T.M. ICF physics; M.J.M. x-ray diagnostics; B.J.M. mode-1 analysis, backscatter; A.J.M. diagnostic management; S.A.M. integrated design physics; A.G.M. x-ray diagnostics; M.M.M. HYDRA code development lead; D.A.M. x-ray diagnostics; E.V.M. x-ray diagnostics; L.M. capsule physics; K.M. gamma diagnostics; N.B.M. hohlraum physics; P.A.M. LPI physics; M.M. optical diagnostics; J.L.M. hohlraum physics; J.D.M. hohlraum physics; A.S.M. neutron diagnostics; J.W.M. hohlraum physics; T.J.M. neutron and gamma diagnostics; K.N. project engineering; J.G.D.N. MOR and PAM stability, SSD improvements, and FC control; A.N. target fab engineering, capsule, and fab planning; R.N. ensembles simulations; M.V.P. HYDRA code development; L.J.P. MOR and PAM stability, SSD improvements, and FC control; J.L.P. ensembles simulations; Y.P. hohlraum physics; B.B.P. hohlraum physics; M. Ratledge capsule fabrication; N.G.R. capsule fabrication; H.R. RTNAD mode-1 analysis; M. Rosen hohlraum physics; M.S.R. x-ray diagnostics; J.D.S. hohlraum physics; J.S. mode-1 analysis; S. Schiaffino capsules; D.J. Schlossberg neutron diagnostics; M.B.S. hohlraum diagnostics; C.R.S. HYDRA code development; H.A.S. NLTE opacities (Cretin) code development; S.M.S. HYDRA code development; K.S. mode-1 metrology;

M.W.S. kinetic physics; S. Shin sagometer data & particle analysis; V.A.S. capsule physics; B.K.S. ensemble simulations; P.T.S. dynamic model, ignition theory; M.S. capsules; S. Stoupin x-ray diagnostics; D.J. Strozzi hohlraum/LPI physics; L.J.S. hohlraum physics; C.A.T. Bigfoot design physics; R.P.J.T. program management; C.T. x-ray framing camera; E.R.T. optical diagnostics; P.L.V. neutron imaging diagnostics; C. R. W. capsule/instability physics; K.W. x-ray diagnostics; C.W. capsule fabrication; C.H.W. neutron diagnostics; B.M.V.W. NIF operations lead; D.T.W. hohlraum physics; B.N.W. project engineering; M.Y. capsule fabrication; S.T.Y. MOR and PAM stability, SSD improvements, and FC control; G.B.Z. computational physics lead.

## Competing interests

The authors declare no competing interests.

## Additional information

J.E. Ralph [1,12] ✉, J. S. Ross [1,12] ✉, A. B. Zylstra [2,12] ✉, A. L. Kritcher [1], H. F. Robey [3], C. V. Young [1], O. A. Hurricane [1], A. Pak [1], D. A. Callahan [4], K. L. Baker [1], D. T. Casey [1], T. Döppner [1], L. Divol [1], M. Hohenberger [1], S. Le Pape [5], P. K. Patel [1], R. Tommasini [1], S. J. Ali [1], P. A. Amendt [1], L. J. Atherton [1], B. Bachmann [1], D. Bailey [1], L. R. Benedetti [1], L. Berzak Hopkins [1], R. Betti [6], S. D. Bhandarkar [1], J. Biener [1], R. M. Bionta [1], N. W. Birge [3], E. J. Bond [1], D. K. Bradley [1], T. Braun [1], T. M. Briggs [1], M. W. Bruhn [1], P. M. Celliers [1], B. Chang [1], T. Chapman [1], H. Chen [1], C. Choate [1], A. R. Christopherson [1], D. S. Clark [1], J. W. Crippen [7], E. L. Dewald [1], T. R. Dittrich [1], M. J. Edwards [1], W. A. Farmer [1], J. E. Field [1], D. Fittinghoff [1], J. Frenje [8], J. Gaffney [1], M. Gatu Johnson [8], S. H. Glenzer [9], G. P. Grim [1], S. Haan [1], K. D. Hahn [1], G. N. Hall [1], B. A. Hammel [1], J. Harte [1], E. Hartouni [1], J. E. Heebner [1], V. J. Hernandez [1], H. W. Herrmann [3], M. C. Herrmann [1], D. E. Hinkel [1], D. D. Ho [1], J. P. Holder [1], W. W. Hsing [1], H. Huang [7], K. D. Humbird [1], N. Izumi [1], L. C. Jarrott [1], J. Jeet [1], O. Jones [1], G. D. Kerbel [1], S. M. Kerr [1], S. F. Khan [1], J. Kilkenny [7], Y. Kim [3], H. Geppert-Kleinrath [3], V. Geppert-Kleinrath [3], C. Kong [7], J. M. Koning [1], J. J. Kroll [1], M. K. G. Kruse [1], B. Kustowski [1], O. L. Landen [1], S. Langer [1], D. Larson [1], N. C. Lemos [1], J. D. Lindl [1], T. Ma [1], M. J. MacDonald [1], B. J. MacGowan [1], A. J. Mackinnon [1], S. A. MacLaren [1], A. G. MacPhee [1], M. M. Marinak [1], D. A. Mariscal [1], E. V. Marley [1], L. Masse [1], K. D. Meaney [3], N. B. Meezan [1], P. A. Michel [1], M. Millot [1], J. L. Milovich [1], J. D. Moody [1], A. S. Moore [1], J. W. Morton [10], T. J. Murphy [3], K. Newman [1], J.-M. G. Di Nicola [1], A. Nikroo [1], R. Nora [1], M. V. Patel [1], L. J. Pelz [1], J. L. Peterson [1], Y. Ping [1], B. B. Pollock [1], M. Ratledge [7], N. G. Rice [7], H. G. Rinderknecht [6], M. Rosen [1], M. S. Rubery [1], J. D. Salmonson [1], J. Sater [1], S. Schiaffino [1], D. J. Schlossberg [1], M. B. Schneider [1], C. R. Schroeder [1], H. A. Scott [1], S. M. Sepke [1], K. Sequoia [7], M. W. Sherlock [1], S. Shin [1], V. A. Smalyuk [1], B. K. Spears [1], P. T. Springer [1], M. Stadermann [1], S. Stoupin [1], D. J. Strozzi [1], L. J. Suter [1], C. A. Thomas [6], R. P. J. Town [1], C. Trosseille [1], E. R. Tubman [1], P. L. Volegov [3], C. R. Weber [1], K. Widmann [1], C. Wild [11], C. H. Wilde [2], B. M. Van Wonterghem [1], D. T. Woods [1], B. N. Woodworth [1], M. Yamaguchi [7], S. T. Yang [1] & G. B. Zimmerman [1]

[1]Lawrence Livermore National Laboratory, P.O. Box 808, Livermore, CA 94551-0808, USA. [2]Pacific Fusion, Fremont, CA 94538, USA. [3]Los Alamos National Laboratory, Mail Stop F663, Los Alamos, NM 87545, USA. [4]Focused Energy, Austin, TX 78758, USA. [5]Laboratoire pour l'utilisation des Lasers Intenses chez École Polytechnique, F-91128 Palaiseau Cedex, France. [6]Laboratory for Laser Energetics, University of Rochester, Rochester, NY 14623, USA. [7]General Atomics, San Diego, CA 92186, USA. [8]Massachusetts Institute of Technology, Cambridge, MA 02139, USA. [9]SLAC National Accelerator Laboratory, Menlo Park, CA 94025, USA. [10]Atomic Weapons Establishment, Aldermaston RG7 4PR, UK. [11]Diamond Materials Gmbh, 79108 Freiburg, Germany. [12]These authors contributed equally: J. E. Ralph, J. S. Ross, A. B. Zylstra. ✉e-mail: ralph5@llnl.gov; ross36@llnl.gov; alex@pacificfusion.com

