## [Peer Review File · Nature Communications]

The impact of low-mode symmetry on inertial fusion energy output in the burning plasma stateREVIEWER COMMENTS

Reviewer #1 (Remarks to the Author):

In this manuscript, the authors find essential parameters that determine the D-T fusion yield under burning plasma conditions and successfully explain the experimental results with a scaling law. Burning plasma refers to a state in which the heating energy of the hot spot by alpha heating exceeds the energy imparted to the hot spot by the imploding shell.

In the pre-burning plasma regime, the reduction in the energy conversion from kinetic energy to the hot spot due to mode-1 asymmetry and the increased radiation loss due to mixing explain the decrease in the experimentally measured yield relative to that predicted by the 1D simulations. Once the plasma enters the burning regime, mode-2 becomes an essential parameter in addition to the above two.

The result presented in this manuscript is outstanding in that it explains the physical quantities necessary for achieving burning and ignition in the inertial confinement fusion (ICF) with experimental results and supporting numerical results and is undoubtedly an essential result in the ICF and plasma fields.

On the other hand, it cannot be concluded that general scaling in ICF has been established as expected by Nature Communications readers, having a broad background from the results and analysis presented in this manuscript.

This Reviewer judges that it would be appropriate to publish the manuscript in a more specialized journal.

The numerical simulations confirm that the sensitivity to $P2/P0$ changes between alpha-on and alpha-off.

However, the authors do not discuss any mechanism for why the $P2/P0$ dependence increases with increasing alpha heating.

Although the authors claim mode-2 is vital in the burning regime, whether this result can apply to the future ignition and high gain regime is still being determined.

If new parameters will be introduced in the future (this Reviewer believes that new parameters will likely be needed), only temporary scaling is obtained in this work, which is insufficient to attract broad interest from readers.

The experimental results and analytical methods are described in a very clear and understandable manner, and the Reviewer has no comments.

This Reviewer's only concern is the unclear scope of the obtained scaling and the general interest in the results.

Since the judgment of general interest is highly dependent on the subjectivity of the reviewers, it is necessary to make a final judgment with consideration of other reviewers' comments.

In the following, I am answering questions from the Nature Communications editorial office,.

* What are the noteworthy results?

The influence of mode-2 was quantitatively demonstrated in ICF by both experiments and numerical simulations in the new plasma regime of burning plasmas, and scaling, including the impact of mode-2, was constructed, showing that a few parameters can reproduce experimental results.

* Will the work be of significance to the field and related fields? How does it compare to the established literature? If the work is not original, please provide relevant references.

This result is significant in the ICF and plasma fields. It is also a completely original result, as only their group can perform this experiment.

* Their conclusions and claims are presented by the experiments, numerical simulations, and model comparisons presented in this paper.

The comparisons among the experiments, numerical simulations, and models justify the authors' conclusions and claims in the current manuscript. On the other hand, at least the applicable range of this scaling needs to be clarified.

* Are there any flaws in the data analysis, interpretation and conclusions? Why is there any flaws in the data analysis, interpretation and conclusions?
There needs to be more discussion as to why the effect of mode-2 appears when alpha heating is on.

* Is the methodology sound?
Yes.

* Is there enough detail provided in the methods for the work to be reproduced?
Yes.

Reviewer #2 (Remarks to the Author):

I recommend this work be published after revision. Below is my report including suggested revisions.
Key Results:

The paper presents a new addition to a postdictive model which modifies the ideal neutron yield for various observed degradations. This new term describes the deleterious effect of mode 2. It is empirical and fit to data. Once included the postdictive model now captures the vast majority of variation in the experimental yield data.

Validity & Analytical approach:

The purely empirical nature of the model may have been a cause for concern, but simulations are used to isolate the effect of mode 2 and show a similar relationship as the data-driven model. The discrepancies between simulation and model are described but not yet fully understood – one possible explanation is raised (P2 swing) which seems reasonable but is not explored further in this work. The fitting of the model is robust (standard least-squares) although I'd like to raise some points/questions for improvement:

1. The P2 correction term should not 1 by definition but this is clearly violated by the fit to I-raum data. Shouldn't $A_j = 1 - B_j$ to prevent this? What is the effect on delta and W if this is done?
2. What is the χ^2 of the new model to the data shown in Fig 4b? The scatter about the line appears smaller than the errorbars suggesting a degree of over-fitting.
3. A footnote says that the data set is down selected for figure 3, is this also true for figure 4? How is the analysis affected if these experiments are left in? What additional 'term' is needed to describe these experiments, why does the mix term not succeed?

Significance:

While the presented empirical fit improves the postdictive model's accuracy, there are a few points worth addressing to improve the work and summarise the significance:

1. The empirical model is fit separately to the I-raum and HYBRID-E data sets. In the supplementary material it is discussed that these data sets are not just repeats of each other. Therefore, at what point are designs significantly diverged that a new empirical fit is needed? While it is clear I-raum and HYBRID-E are different, how about different designs within a single hohlraum design philosophy? This somewhat limits the model's ability to act as a P2 correction term in the generic ICF burning plasma state.
2. Following on from this point, it is clear from the simulations that this P2 correction term is sensitive to the degree of alpha heating. Therefore, the empirical fit coefficients are in fact also a function of yield amplification. What yield amplification range are these coefficients suitable for? Is the empirical model expected to work in the ignition regime? A sentence at the end of the paper suggests that the P2 sensitivity will decrease in the future, suggesting limited scope of the empirical model presented.
3. While it is clear that the combination of models describes the degradation of yield effectively (for the down-selected data set). It is unclear to me how this information is used. Is it triage, i.e. whichever eta parameter is smallest is our biggest problem? How effectively can remedies be applied

given shot-to-shot variation?

Data and methodology:

The data and methodology is well presented and it is clear what steps have been taken in the analysis.

It is worth addressing how the X-ray P2/P0 correlates with other measures of mode 2: neutron imaging and Tion asymmetry?

Clarity and context:

The description of the experiments and models are clear. This paper requires heavily on past work and sufficient explanation and referencing is given for this.

Dear Reviewers,

On behalf of the experimental collaboration working on experiments at the National Ignition Facility at Lawrence Livermore National Laboratory, we would like to thank you for finding the time to review and consider this manuscript for publication in Nature Communications. We have made several important changes based on your suggestions and comments which we believe has resulted in a stronger manuscript.

In addressing these comments, we would like to resubmit this revised manuscript to Nature Communications for publication. The work described in this analysis has allowed the team at Lawrence Livermore to focus on addressing the three primary degradations found in this work to have the greatest impact on shot-to-shot performance variability and by noting the very high sensitivity to mode-2 in particular. This understanding has helped the team make rapid progress.

A full report follows in which we address each point individually.

Sincerely,
Joseph E. Ralph Ph. D.
Lawrence Livermore National Laboratory

REVIEWER COMMENTS

Reviewer #1 (Remarks to the Author):

In this manuscript, the authors find essential parameters that determine the D-T fusion yield under burning plasma conditions and successfully explain the experimental results with a scaling law.

Burning plasma refers to a state in which the heating energy of the hot spot by alpha heating exceeds the energy imparted to the hot spot by the imploding shell.

In the pre-burning plasma regime, the reduction in the energy conversion from kinetic energy to the hot spot due to mode-1 asymmetry and the increased radiation loss due to mixing explain the decrease in the experimentally measured yield relative to that predicted by the 1D simulations.

Once the plasma enters the burning regime, mode-2 becomes an essential parameter in addition to the above two.

The result presented in this manuscript is outstanding in that it explains the physical quantities necessary for achieving burning and ignition in the inertial confinement fusion (ICF) with experimental results and supporting numerical results and is undoubtedly an essential result in the ICF and plasma fields.

On the other hand, it cannot be concluded that general scaling in ICF has been established as expected by Nature Communications readers, having a broad background from the results and analysis presented in this manuscript. This Reviewer judges that it would be appropriate to publish the manuscript in a more specialized journal.

This manuscript describes the essential understanding that was required to achieve a burning plasma which directly led to ignition and target energy gain. Prior general scaling's have been made [i.e. Lindl [4], R. Betti PRL 114, 255003 (2015)] in theory articles. This is the first experimentally validated scaling that informs how and why a burning plasma was achieved in the laboratory. As discussed in more detail below, this work is directly relevant to achieving ignition. Additionally, the yield scaling with degradations can directly inform prior generalized models of ignition. We modify the text to highlight these points.

To address the reviewer's comment on more clearly making the connection to a general finding we have added the following text starting on line 20,

“Harnessing and controlling fusion reactions for use as an energy source remains a major scientific objective with far-reaching potential. At the time of this writing, experiments at the National Ignition Facility (NIF) using the indirect-drive inertial confinement fusion (ICF) approach^{3,4}, have made significant progress in that pursuit⁵. *These advances were directly enabled by using experimental observations to identify and mitigate degrading factors that reduce the rate of fusion self-heating and prevent ignition. Generalized ignition criteria often cast ignition in terms of an amplification of in-cent coupled energy, pressure, and yield [J. Lindl et. al Phys. Plasmas 25 122704 2018, R. Betti et. al. PRL 114, 255003 (2015)]. The formalism developed in this work informs these criteria in terms of how compressional asymmetries and enhanced radiative loss impact the fusion yield amplification in a regime with significant fusion self-heating and directly relevant to ignition.*”

More detail:

Response Fig. 1: Time integrated and spatially resolved neutron emission images. In each image, the fusion yield is displayed in the upper left, and experiment month and year displayed in the lower left.

Response Fig. 1 shows the evolution of the neutron emission from the burning plasma regime discussed in this paper (N210307), shown in a), to the first experiment to achieve ignition shown in d). From March of 2021 to August of 2021, improvements to the target quality, radiation drive and implosion symmetry were made. An additional increase in laser energy and resulting radiation drive as well as an increase in capsule mass was made going from August of 2021 to September of 2022 shown in c). As can be seen, this resulted yield comparable to the August 2021 experiment, but with an oblate hot spot with a P2/P0 asymmetry of -20%. The framework developed in this manuscript would suggest that, with all other degradations being equal, fully correcting the asymmetry of the Hybrid E type implosion should result in a ~2.5X increase in fusion yield. In December of 2022 a symmetry adjustment was made, and the fusion yield increased 2.6X consistent with expectations given the relative change in the mode 1 and mix degradations. Without this study, quantifying the expected and measured impact of the symmetry change in the presence of other degradations would not be possible.

For a fixed mass of DT fuel, as in the experiments discussed here, the $ITFX_{no \alpha}$ can be written as

$$\propto ()^{2.1}$$

The $Y_{no \alpha}$ of the system can be rewritten as, $Y_{no \alpha} \propto n_{DT}^2 \langle \sigma v \rangle V \tau \propto E_h s T_i^{1.5} P \tau$ given that the expected temperature in the absence of alpha-heating on the cusp of ignition is of 4.3 keV where the DT reactivity $\langle \sigma v \rangle \propto T_i^{3.5}$. Here, $E_h s$, is the energy coupled to the hot spot, T_i is the plasma ion temperature, P is the plasma pressure and τ is the duration over which the burn rate can be sustained. Asymmetries in compression from mode 1 and mode 2

asymmetries, or increased radiative loss from higher atomic number mix, can reduce the energy coupled to the hot spot, Ehs, and reduce the temperature and pressure, lowering the Yno α , the IIFXno α and the Yamp.

Therefore, understanding how the incident yield varies with degradations can inform and be coupled into a generalized theory for ignition. In the work described in this manuscript the degradation to the fusion yield is described for systems with alpha-particle heating and can be incorporated into the IIFX criteria with alpha heating (see P. Patel et al, Phys. Plasma (2020) for relating IIFX to the IIFXno α) and to Yamp. The individual degradation factors for mode 1, mix and mode 2 asymmetry correspond to changes in the yield amplification associated with changes to the generalized IIFX criteria.

$$(-\ln(\))^{2.13} = \frac{0.9}{0.9} = \frac{0.9^{0.47}}{0.9^{0.47}} = 1 - \frac{\Delta}{0.9} = 1 - \frac{\Delta}{0.9}$$

Here f represents the fractional change in the ignition threshold factor that is associated with a particular degradation factor h.

The numerical simulations confirm the sensitivity to P2/P0 changes between alpha-on and alpha-off. However, the authors do not discuss any mechanism for why the P2/P0 dependence increases with increasing alpha heating.

This is a good point. The following sentences have been added to the conclusion to clarify this point.

“From this analysis, we find that these experiments would likely have entered the burning plasma regime, had the P2 symmetry been optimized. This high sensitivity to P2 is reinforced by simulation results which indicate that alpha heating, E α , greatly increases the sensitivity of P2. Under burning plasma conditions alpha-heating heats the hot core more than compressional work (E α > EPdV) leading to higher fusion energy output[1] . The data analysis and simulation results presented indicate that the rapid alpha heating process has been suppressed by degradation from asymmetric P2. The rapid increase in performance that occurs as the implosion symmetry is improved with increased alpha heating significantly increases the performance sensitivity to P2 in experiments otherwise meeting the threshold for burning plasma conditions.”

Although the authors claim mode-2 is vital in the burning regime, whether this result can apply to the future ignition and high gain regime is still being determined.

As discussed above and shown in Response Fig. 1, in recent experiments on the NIF, a gain of greater than 1 was achieved. Simulations and experimental evidence, based in large part on this work, indicated that a P2 correction was required to achieve gain > 1 and that the increase in measured performance is consistent with these results.

If new parameters will be introduced in the future (this Reviewer believes that new parameters will likely be needed), only temporary scaling is obtained in this work, which is insufficient to attract broad interest from readers.

In this response, we have shown how to generalize the impact of perturbations to an ignition theory. As new limiting degradations are discovered with future experiments, their impact can be incorporated into this framework. We emphasize that what has been developed is not a temporary scaling, but one that accounts, simultaneously, for all the currently known principal degradation mechanisms that we can diagnose. It can currently describe the relative behavior of our experiments. As these top-level degradations are addressed, we fully expect to uncover new limiting degradations – however, without this work, it would be impossible to account for the 2nd order or next level limiting degradations. This approach should be of general interest to many fields

which work on highly coupled problems as it offers a way to untangle and quantify multiple effects to gain understanding.

The experimental results and analytical methods are described in a very clear and understandable manner, and the Reviewer has no comments.

This Reviewer's only concern is the unclear scope of the obtained scaling and the general interest in the results.

Since the judgment of general interest is highly dependent on the subjectivity of the reviewers, it is necessary to make a final judgment with consideration of other reviewers' comments.

The reviewer has acknowledged, "In this manuscript, the authors find essential parameters that determine the D-T fusion yield under burning plasma conditions and successfully explain the experimental results with a scaling law." Since this is the first set of experiments where these burning plasma conditions have been achieved and studied leading to the scaling presented here, we believe this work is currently and will remain of general interest.

In the following, I am answering questions from the Nature Communications editorial office.,

* What are the noteworthy results?

The influence of mode-2 was quantitatively demonstrated in ICF by both experiments and numerical simulations in the new plasma regime of burning plasmas, and scaling, including the impact of mode-2, was constructed, showing that a few parameters can reproduce experimental results.

* Will the work be of significance to the field and related fields? How does it compare to the established literature? If the work is not original, please provide relevant references.

This result is significant in the ICF and plasma fields. It is also a completely original result, as only their group can perform this experiment.

* Their conclusions and claims are presented by the experiments, numerical simulations, and model comparisons presented in this paper.

The comparisons among the experiments, numerical simulations, and models justify the authors' conclusions and claims in the current manuscript. On the other hand, at least the applicable range of this scaling needs to be clarified.

As discussed above and now in the manuscript, we are still finding good agreement for P2 sensitivity even in igniting and near igniting plasmas. Also discussed, these results can be incorporated into a generalized ignition criteria such that we can quantify their impact across a wide range of incident conditions. However, [Kritcher et. al Phys. 585 Plasmas 21, 042708 (2014)] shows this may be near the threshold where saturation or robust burning/igniting occurs.

* Are there any flaws in the data analysis, interpretation, and conclusions? Why is there any flaws in the data analysis, interpretation and conclusions?

There needs to be more discussion as to why the effect of mode-2 appears when alpha-heating is on.

* Is the methodology sound?

Yes.

* Is there enough detail provided in the methods for the work to be reproduced?

Yes.

Reviewer #2 (Remarks to the Author):

I recommend this work be published after revision. Below is my report including suggested revisions.

Key Results:

The paper presents a new addition to a postdictive model which modifies the ideal neutron yield for various observed degradations. This new term describes the deleterious effect of mode 2. It is empirical and fit to data. Once included the postdictive model now captures the vast majority of variation in the experimental yield data.

Validity & Analytical approach:

The purely empirical nature of the model may have been a cause for concern, but simulations are used to isolate the effect of mode 2 and show a similar relationship as the data-driven model. The discrepancies between simulation and model are described but not yet fully understood – one possible explanation is raised (P2 swing) which seems reasonable but is not explored further in this work.

The fitting of the model is robust (standard least-squares) although I'd like to raise some points/questions for improvement:

1. The P2 correction term should not [greater than] 1 by definition but this is clearly violated by the fit to I-raum data. Shouldn't $A_j = 1 - B_j$ to prevent this? What is the effect on delta and W if this is done?

This is a good suggestion. We have changed the method of normalization of both the analyzed data and model fit to accommodate this suggestion. We now enforce a maximum η_{P2} of 1, so that $A_j + B_j = 1$. Because both data and model are renormalized, the width (W) and offset (delta_j) remain unchanged. These changes are reflected in Figs. 3 (c) and (d) as well as Tables 1 and 2 in the appendix.

NEW FIGURES (3c and 3d)

Table 2 | Fit parameter values used for η_{P_2} .

	HYBRID-E	I-Raum
A_j	0.9225	0.9502
δ	0.1180	0.1180
W_j	0.3058	0.1820
B_j	0.0775	0.0498

Furthermore, we have described the changes in the paragraph starting on line 303. The text in paragraphs starting on lines 314 and 319 have also been slightly rearranged to bring the description of the model fit earlier. Because of this change however, and due to the limited simulation data available for the I-Raum, we have assumed (for the case of the I-Raum) that the maximum for the simulation results occurs at $P_2/P_0 = 0$ which follows the slope of the experimental data but is offset in the oblate direction similar to the HYBRID-E simulations. This change required a second x-axis for the I-Raum simulation results, making the axes (top sim and bottom experiment) for the I-Raum plot similar to that of HYBRID-E. This change is reflected in the caption of Fig. 3, “ Again, to compare the sensitivity between the simulations and experiments, the P_2/P_0 axes for the simulations results (labeled above the plot) have required shifting relative to the experimental P_2/P_0 axes (shown below) because of a systematic offset.”

The result of these changes is a much more useful empirical model that is now analogous to the correction terms for mix and mode-1. For the reader, with the y-axes the same, the reader can now more easily consider and compare the impact of P_2 relative to the other degradation mechanisms.

The choice of normalization for the model is arbitrary. In the prior method, the fit and data analysis was normalized to the highest performing experiment for each data set (HYBRID-E and I-Raum) which allowed the empirical model to exhibit values greater than 1. This is particularly true for the I-Raum case, as pointed out. However, since the final models for I-Raum and HYBRID-E require differing proportionality constants to fit equation (1), the normalization factor of the P_2 scaling could be renormalized without impacting the primary result.

2. What is the χ^2 of the new model to the data shown in Fig 4b? The scatter about the line appears smaller than the errorbars suggesting a degree of over-fitting.

The χ^2 for the HYBRID-E model including the P_2 correction using the 4-parameter fit 0.27, from fit to data. For the record, it is possible to fit the data with only 3 -parameters (omitting B), resulting in a χ^2 (best fit) of 0.31, still a very good fit.

However, to try to capture relevant patterns in the data and simulations, we chose the 4-terms given in the P_2 fit. The terms, amplitude (A), offset (δ), width (w) terms as well as a term for the plateau (B). The amplitude is just a normalization term, and now has been adjusted so that $A+B=1$. Furthermore, there is a width indicating the performance sensitivity to P_2/P_0 . Finally, we measure an offset(δ) relative to $P_2/P_0=0$ outside the error bars that may result from conduction which is related to surface-to-volume ratio or from a minimization of residual kinetic energy (RKE)lost from P_2 swinging with time. Simulations with different hohlraums and models also show a variation of offset. Finally, simulations and experiment do show consistently some performance ($Y_n > 0$ and $T_{ion} > 0$) even for implosions with P_2/P_0 far from the point of maximum performance. The plateau “B” is seen in both experiment and simulation. As the implosion becomes more and more oblate with $P_2/P_0 < -0.5$ for example, we always measure some yield and T_{ion} and this is shown to be expected from simulation. Likewise, as the P_2/P_0 increases, the yield does not go to zero, and is also not expected to. This can be seen in Figs. 3 (d) the I-Raum simulation with $Y_n > 0$ for very high P_2/P_0 . Indeed, in both Figs.3(c) and (d), this is more easily seen in the “no-alpha heating” simulations which fall off much more slowly with P_2 . This is noted on line 329, “ A minimum performance level, B_j , is allowed because we find measurable neutron yields in experiments, even with severe asymmetry.” In addition, this “B” term limits the interpretation of low performing implosions that are very P_2 asymmetric, from overestimating the potential gains from a P_2 correction.

3. A footnote says that the data set is down selected for figure 3, is this also true for figure 4? How is the analysis affected if these experiments are left in? What additional 'term' is needed to describe these experiments, why does the mix term not succeed?

Yes, both figures 3 and 4 use the down selected data. A total of 5 HYBRID-E DT experiments were omitted from this analysis. All I-Raum data is included.

To be more explicit, the footnote has been updated, "Data from the three HYBRID-E experiments with very poor-quality DT ice layers have been omitted from all analysis presented in this article..."

Below are plots of the HYBRID-E data. In the plot below left, we can see that these points are well below the trend. Three of these were omitted as you said because of ice layer (circled in green) and, one of these was omitted because of very large particles that caused jets. The ice layers from the green points are circled on the right plot where layer quality is worse as points move up and to the right. These points as well as the data with large coalescing particle detections may result in very large perturbations to the implosion. Measuring radiative mix assumes the hot spot losses are from x-ray emission, and typically work for the case of mix that occurs as the hot core heats up. Very large perturbations that occur early in the implosion, can lead to early mix that would reduce temperatures so significantly that x-ray emission would be suppressed. This degradation could show up in so-called cold mix (presently a process for quantifying this mix is being developed) which can help determine losses from non-radiative mixing.

Another possibility, that may occur in these large perturbation cases, is the development of aneurisms, as discussed in Hurricane et. al., which would add another loss term to the power balance equation, in which the DT gas in the core is no longer well confined and can leak out of the hot spot taking energy with it.

Finally, the last point (included for completeness) is lower because the assumed adiabat for all cases is 3 (by design), and the design adiabat for that experiment was 4. In fact, using 4 does bring the data point in-line with others. However, determination of the adiabat, as mentioned in the manuscript requires radiation hydrodynamic simulations, and for an isolated point, using simulations tuned to that point leads to a circular argument which is not the case if the adiabat (shock merger depth) is constant.

Significance:

While the presented empirical fit improves the postdictive model's accuracy, there are a few points worth addressing to improve the work and summarize the significance:

1. The empirical model is fit separately to the I-Raum and HYBRID-E data sets. In the supplementary material it is discussed that these data sets are not just repeats of each other. Therefore, at what point are designs significantly diverged that a new empirical fit is needed?

Differences in P2 symmetry degradations may arise from the different time dependent radiation flux asymmetries experienced which depend on the dynamics of the imploding shell as well as the hohlraum. In all of our current implosions, the P2 asymmetry is largely driven by the ingress of the Au bubble and CBET as discussed in the text. The difference between the P2 symmetry impact for the two sets of data are likely due to the differences in the shape of the cylindrical and I-Raum radiation cavities and differences in the required CBET. These differences as well as differences in laser pulse shapes contribute to the differing time dependent P2 radiation flux asymmetry which drive different shell mode-2 swings.

The figure below is a direct comparison of η_{P2} from the two data sets (HYBRID-E blue, I-Raum red) showing a similar but clear offset that may be a result of swings or other physics.

We plan to try to isolate the mode-2 swings and their impact in future experiments.

While it is clear I-Raum and HYBRID-E are different, how about different designs within a single hohlraum design philosophy? This somewhat limits the model's ability to act as a P2 correction term in the generic ICF burning plasma state.

Assuming all other etas (mix and mode-1) = 1, the sensitivity of P2 should be a function of the maximum yield amplification (alpha heating), as you note below. Understanding what design differences impact mode sensitivity is certainly of great interest and certainly warrants its own detailed study. The analysis of this paper suggests a hypothesis that cylindrically symmetric hohlraums may be represented by a gaussian form and that swings or other physics may primarily influence the width and offset.

2. Following on from this point, it is clear from the simulations that this P2 correction term is sensitive to the degree of alpha heating. Therefore, the empirical fit coefficients are in fact also a function of yield amplification. What yield amplification range are these coefficients suitable for? Is the empirical model expected to work in the ignition regime? A sentence at the end of the paper suggests that the P2 sensitivity will decrease in the future, suggesting limited scope of the empirical model presented.

Although, the framework of the model was developed for yields consistent with experiments on the verge of burning plasma [Hurricane et. al. 61 014033 PLASMA PHYSICS AND CONTROLLED FUSION 2019], simulations with yields exceeding $2e18$ (>4 MJ fusion yield) still produce sensitivities consistent with these results. Furthermore, as discussed above we are seeing this sensitivity in ignition experiments ($Y_{amp} > 30$). Though we expect, very robust alpha heating may eventually dominate any implosion. To clarify this point, we have modified the last sentence starting at line 436, "As experiments improve, simulations do indicate that gains from E_{α} will increase faster than degradations and a larger fraction of the fuel will be burned up. As we make these improvements to the 1D parameters, the sensitivity of performance to $P2/P0$ and other degradations is expected to plateau as the fuel robustly burns^[12]. "

As performance increases, $P2$ sensitivity is *expected* to decrease forming a wider region where $P2$ sensitivity is flat. Note that we have not yet observed this in experiments on the NIF even with ignition and gain.

3. While it is clear that the combination of models describes the degradation of yield effectively (for the down-selected data set). It is unclear to me how this information is used. Is it triage, i.e. whichever eta parameter is smallest is our biggest problem? How effectively can remedies be applied given shot-to-shot variation?

The short answer is yes, it is a triage but the inputs contributing to these degradations are mostly independent. This allows us to address them independently. However, to test the efficacy of any change, we need to be able to assess whether or not the experiment was degraded and how. We can then address each degradation, tailoring the laser uniformity and hohlraum design to address mode-1, mix, by optimizing target quality and design stability, and $P2$ with changes such as increased CBET, decreased picket energy, or a shorter pulse duration. This approach has allowed us to navigate the myriad of potential design improvements more rapidly and with more certainty than ever before and has helped lead to ignition. It is also informing us to the limits of new designs and new limiting degradations.

Data and methodology:

The data and methodology is well presented and it is clear what steps have been taken in the analysis.

It is worth addressing how the X-ray $P2/P0$ correlates with other measures of mode 2: neutron imaging and T_{ion} asymmetry?

The quality, robustness and consistency of analysis of x-ray images was very high over the entire period of these experiments which is why we chose to rely on that data analysis. That said, there is a strong correlation between neutron mode-2 analysis (measured as a $Y_{lm}(2,0)$ fit to a 3D reconstruction) and x-ray mode-2 (Legendre fit) that reveals a similar trend, see correlation between plot below. As far as T_{ion} distribution, on this series of experiments, we do not find a correlation between $P2/P0$ and T_{ion} asymmetry. Please see plots below.

Clarity and context:

The description of the experiments and models are clear. This paper requires (relies) heavily on past work and sufficient explanation and referencing is given for this.

REVIEWERS' COMMENTS

Reviewer #1 (Remarks to the Author):

This reviewer is satisfied with the author's response and is now convinced of the importance of this study. This revised manuscript is worthy of publication in Nature Communications.

Reviewer #2 (Remarks to the Author):

I thank the authors for their detailed reply to my review. I believe the revised manuscript is now fit for publication.